# Alpha-herpesvirus US1 interacts with cGAS to suppress type I IFN responses and antiviral defense

Weiyu Qu[1]☯, Ye Yuan[1]☯, Dongdong Shen[2]☯, Jiwen Zhang[2]☯, Xiangyu Huang[1]☯, Lei Wu[1], Hongyan Yin[1], Zhenchao Zhao[1], Haiwei Wang[2], Lvye Chai[1], Jiayang Wu[3], Xijun He[2], Cheng Zhu[3], Dongming Zhao[2]*, Xin Li [1]*

**1** State Key Laboratory of Veterinary Public Health and Safety, College of Veterinary Medicine, China Agricultural University, Beijing, China, **2** State Key Laboratory of Animal Disease Control and Prevention, Harbin Veterinary Research Institute, Chinese Academy of Agricultural Sciences, Harbin, China, **3** Tianjin Key Laboratory of Function and Application of Biological Macromolecular Structures, School of Life Sciences, Tianjin University, Tianjin, China

☯ These authors contributed equally to this work.
* zhaodongming@caas.cn (DZ); xinli2021@cau.edu.cn (XL)

## Abstract

Alpha-herpesviruses, including pseudorabies virus (PRV) and herpes simplex virus type 1 (HSV-1), cause severe diseases in a wide range of hosts. However, the precise mechanisms of immune evasion by alpha-herpesviruses remains elusive, hindering the development of broad-spectrum antiviral vaccines and drugs. Here, we demonstrate that the immediate early protein US1, encoded by alpha-herpesviruses, directly interacts with cGAS, suppressing its dsDNA binding and enzymatic activity. Structural analysis using AlphaFold reveals a conserved overlapping region within PRV and HSV-1 US1 proteins. Deletion of these peptides leads to increased cGAS-mediated IFN-β production. Meanwhile, both synthetic and purified SUMO-fused US1 peptides significantly inhibit cGAS activity across species, with the SUMO-fused US1 peptides directly binding to the catalytic domain of cGAS. Both US1-deficient viruses (PRV-ΔUS1 and HSV-1-ΔUS1) exhibit higher IFN-β production and enhanced signaling through the cGAS-STING pathway. Importantly, mice infected with PRV-ΔUS1 or HSV-1-ΔUS1 show increased IFN-β secretion and reduced viral loads. In conclusion, overlapping peptides from US1 protein of alpha-herpesviruses antagonize cGAS-mediated innate immune responses, highlighting a promising target for the development of broad-spectrum inhibitors to counteract herpesvirus infections.

**Data availability statement:** All data are available in the main text or the Supporting information.

**Funding:** This study was supported by the National Natural Science Foundation of China (32373020) and National Key Research & Development Program of China (2022YFD1800300) to X.L. and National Natural Science Foundation of China (22077094, 22007071) to C.Z.The funders had no role in study design, data collection and analysis, decision to publish, or preparation of the manuscript.

**Competing interests:** The authors have declared that no competing interests exist.

## Author summary

Type I interferon (IFN-I) restricts viral replication, and cGAS activates its production upon detecting viral DNA. Alpha-herpesviruses, such as pseudorabies virus (PRV) and herpes simplex virus type 1 (HSV-1), cause severe neurological and mucocutaneous problems in their hosts. However, the unified mechanisms behind immune evasion in alpha-herpesviruses remain largely unknown, hindering the development of broad-spectrum vaccines and antiviral drugs. Here, we demonstrate that overlapping peptides from alpha-herpesvirus-encoded immediate early US1 proteins directly bind to and inhibit cGAS activation. Deletion of these peptides in US1 proteins resulted in enhanced cGAS-mediated IFN-β production. Importantly, PRV-ΔUS1 or HSV-1-ΔUS1 exhibited increased IFN-β secretion and reduced viral loads in vivo. This potential mechanism reveals a unique strategy for developing broad-spectrum antiviral drugs targeting alpha-herpesviruses.

## Introduction

Innate immune DNA sensors, such as cyclic GMP-AMP synthase (cGAS) [1], detect double-stranded DNA (dsDNA) from pathogenic invaders or damaged mitochondria, triggering host immune responses. Upon recognizing dsDNA, cGAS oligomerizes and is activated to catalyze the synthesis of 2′3′-cGAMP, which subsequently binds to and activates stimulator of interferon genes (STING) [2,3]. Once activated, STING initiates a signaling cascade by activating TANK binding kinase 1 (TBK1), which phosphorylates and activates interferon regulatory factor 3 (IRF3), leading to the induction of type I interferons (IFN-I) [4,5].

Alpha-herpesviruses, including PRV, HSV-1 and Varicella-Zoster Virus (VZV), pose significant threats to public health. PRV, an important alpha-herpesvirus, causes acute infectious diseases characterized by fever and encephalomyelitis [6] in its natural host pigs, and it also poses a risk of cross-species transmission, including to humans [7]. In humans, PRV infection can present with acute onset of fever, headache, seizures, and consciousness disorders. Notably, the Chen group successfully isolated a PRV strain from the cerebrospinal fluid of a patient [8], highlighting the virus's ability to cross species barriers and its potential threat to human health, warranting close attention. Among these viruses, HSV-1 is notably prevalent, causing a range of symptoms in humans, such as herpes labialis, corneal conjunctivitis, and genital infections [6]. Additionally, HSV-1 encephalitis (HSE) results in severe neuroinflammation and neurological impairment in the central nervous system (CNS), manifesting in a wide array of clinical symptoms, including cognitive dysfunction, personality changes, aphasia, and seizures [9]. Moreover, VZV causes chickenpox and shingles, with the latter particularly inducing prolonged neuralgia in the elderly, significantly impairing quality of life [10]. The widespread occurrence of alpha-herpesviruses contributes to devastating diseases and substantial economic losses globally.

Previous studies have shown that herpesviruses-encoded proteins mediate the evasion of innate immune defenses by targeting cGAS and inhibiting its activity. This has been documented with several viral proteins, including UL21 of PRV [11]; UL21, VP22, UL37, UL41 and UL56 of HSV-1 [11–15]; UL31, UL42, and UL83 of human cytomegalovirus (HCMV) [16–18]; ORF52 and ORF73 of Kaposi's sarcoma associated herpes virus (KSHV) [19–22]; and ORF9 of VZV [23]. UL21 [11] antagonizes IFN-β production by promoting cGAS degradation through the autophagolysosomal pathway, while VP22 [24] inhibits cGAS by disrupting cGAS-DNA phase separation. Despite alpha-herpesviruses encoding between 60–100 viral proteins, the inhibitory mechanisms of these viral proteins remain poorly understood. Furthermore, it is unclear if a common strategy exists across herpesviruses for antagonizing innate immunity, particularly concerning the structural homology of these viral proteins.

US1, a viral immediate early protein, also known as ICP22, is a multifunctional protein modified post-translationally [25], playing diverse roles in the herpesvirus life cycle. It contributes to viral latent infection [26], regulates viral gene transcription [27], and aids in viral particle maturation [28]. In addition, US1 is involved in various cellular processes, including apoptosis, autophagy, and antiviral responses [29]. However, the impact of US1 on innate immune regulation remains largely unexplored. While extensive research has focused on US1 in human herpesviruses, there is a gap in understanding the function of PRV US1 and whether a common strategy exists among alpha-herpesviruses' US1 proteins in evading innate immune restrictions.

In this study, we demonstrate that alpha-herpesviruses US1 proteins inhibit cGAS-mediated activation through a combination of cellular, biochemical and murine studies. We observed that US1 significantly reduced cGAS-mediated IFN-I production by directly interacting with cGAS, thereby disrupting its DNA binding and enzymatic activity. Notably, structural alignment revealed that two overlapping peptides from PRV and HSV-1 US1 were crucial for US1-mediated inhibition of cGAS. Synthetic peptides derived from US1 also inhibited cGAS activity. In addition, SUMO-fused US1 peptides directly interacted with the catalytic domain of cGAS, suggesting potential targets for drug development. These findings uncover a conserved strategy by which alpha-herpesviruses counteract host innate immunity, utilizing the structurally conserved region of US1 to antagonize cGAS activity, thereby illustrating a sophisticated mechanism of immune evasion.

## Results

### Alpha-herpesvirus US1 inhibits cGAS-mediated IFN-β induction

Upon viral infection, cGAS is activated, initiating a cascade of signaling events that lead to IFN-I production [1–5]. However, we observed that alpha-herpesviruses, particularly PRV and HSV-1, failed to induce IFN-β in PK-15 and HeLa cells, suggesting the presence of mechanisms that suppress IFN-I induction by these viruses (Fig 1A and 1B). To identify PRV-encoded proteins responsible for suppressing the activation of the cGAS-STING pathway, we cloned genes encoding viral proteins US1, US2, US3, US7, US9, UL11, UL31 and EP0, and evaluated their effects on IFN-β promoter activation in HEK-293T cells using a dual-luciferase reporter assay (Fig 1C). In addition to the known inhibitory effects of US3 and EP0 [30,31], we observed that US1 significantly inhibited cGAS-STING-mediated IFN-β reporter activation. To rule out false-positive results, we verified the effect of PRV US1 on IFN-β regulation by sequentially increasing the dose of US1 during transfection. PRV US1 progressively reduced cGAS-STING-triggered activation of the IFN-β promoter in a dose-dependent manner (Fig 1D). Consistently, qPCR analysis showed that PRV US1 impaired IFN-β expression in immortalized porcine alveolar macrophages (PAM-Tang) cells upon poly (dA:dT) stimulation (Fig 1E) and in HEK-293T cells following transfection of porcine cGAS (pcGAS) and porcine STING (pSTING) (Fig 1F). Additionally, PRV US1 expression reduced the mRNA levels of *ISG15*, *ISG54* and *ISG56* in HEK-293T cells transfected with pcGAS or pSTING (S1A-S1C Fig). We further analyzed the amino acid sequences of US1 proteins from the alpha-herpesvirus family and constructed a neighbor-joining (NJ) tree using MEGA software (Fig 1G). This analysis revealed a surprising low sequence homology (15%) between PRV and HSV-1 US1 proteins (S2 Fig). Despite the low sequence homology, both PRV and HSV-1 US1 inhibited human cGAS-STING mediated IFN-β promoter activity (Fig 1H and 1I). To investigate how

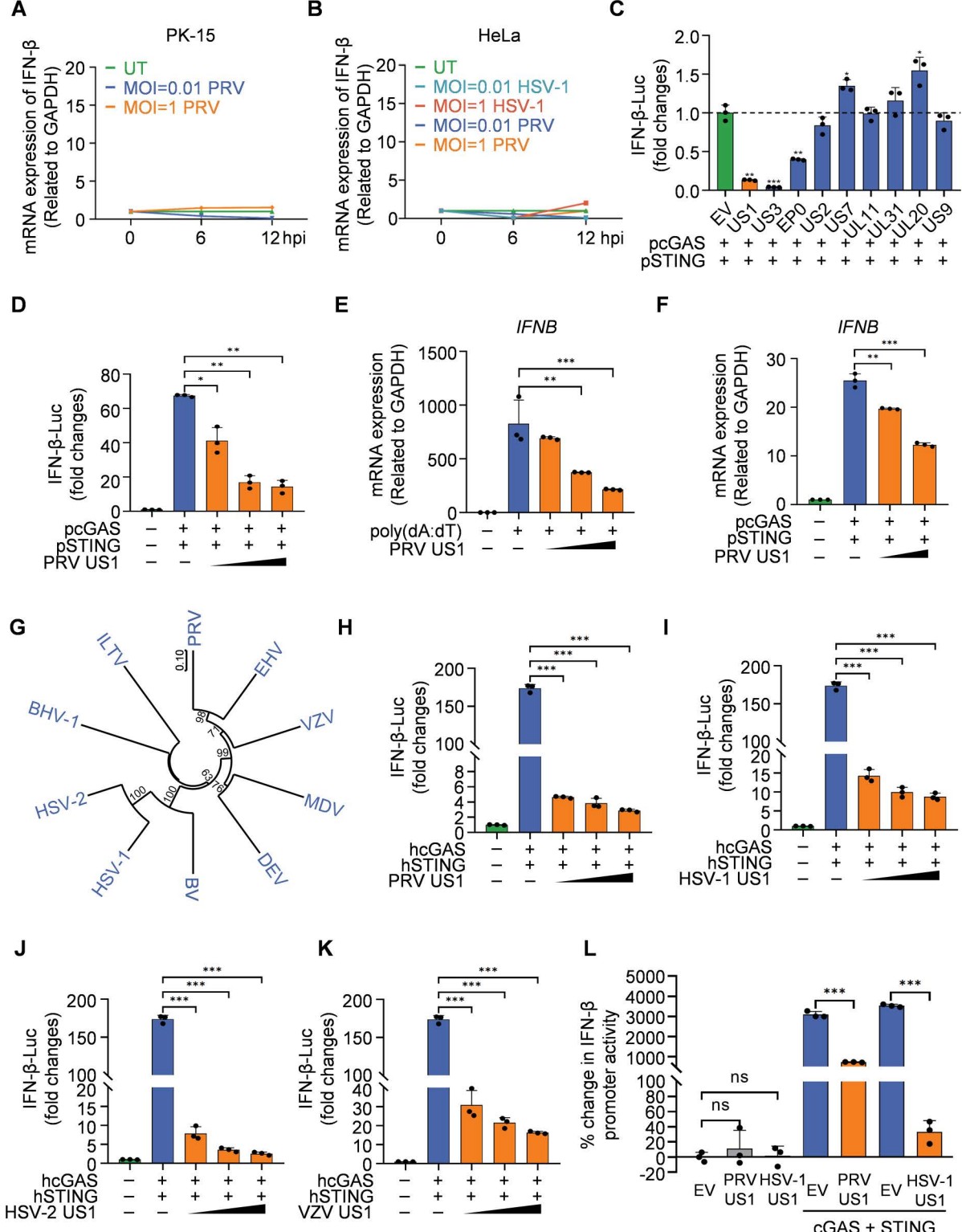

**Fig 1. Alpha-herpesvirus US1 protein inhibits cGAS-mediated IFN-β induction.** (**A, B**) qPCR analysis of IFN-β mRNA expression in PK-15 cells infected with PRV (**A**) and HeLa cells infected with PRV or HSV-1 (**B**). Cells were infected with PRV or HSV-1 at 0.01 or 1 plaque forming unit (PFU)/

well and harvested at the indicated times. **(C)** Detection of IFN-β promoter activation in HEK-293T cells transfected with IFN-β-Luc reporter, expression plasmids encoding pcGAS, pSTING and PRV proteins or empty vector for 24 hours using dual-luciferase reporter assay. The expression levels of US1 ($P = 0.0001$), US3 ($P < 0.0001$), and EP0 ($P = 0.0005$) were significantly downregulated, whereas US7 ($P = 0.0087$) and UL20 ($P = 0.0089$) exhibited mild upregulation. **(D)** HEK-293T cells were treated as in **(C)**, with transfections using increasing doses of PRV US1. Control vs low-dose US1 ($P = 0.0041$), control vs medium-dose US1 ($P < 0.0001$), control vs high-dose US1 ($P < 0.0001$). **(E)** PAM-Tang cells were transfected with increasing doses of PRV US1. At 24 hpt, cells were stimulated with poly(dA:dT) at a concentration of 1 ug/mL. After another 24 h, cells were harvested for RNA extraction and RT-PCR detection of IFN-β. Control vs medium-dose US1 ($P = 0.0227$), control vs high-dose US1 ($P = 0.0082$). **(F)** HEK-293T cells were treated similarly to those in **(D)**, but without IFN-β-Luc transfection. The cells were harvested for RNA extraction and RT-PCR detection of *IFNB* and *GAPDH* mRNA levels. Control vs low-dose US1 ($P = 0.0021$), control vs high-dose US1 ($P < 0.0001$). **(G)** The multiple sequence alignment analysis of US1 proteins from alpha herpesviruses using MEGA software. **(H)** Detection of IFN-β promoter activation in HEK-293T cells transfected with IFN-β-Luc reporter, expression plasmids encoding hcGAS, hSTING and PRV US1 with increasing doses or empty vector for 24 hours using dual-luciferase reporter assay. ***$P < 0.0001$. **(I-K)** HEK-293T cells were treated as in **(H)**, but HSV-1 US1 **(I)**, HSV-2 US1 **(J)** or VZV US1**(K)** was used instead of PRV US1. ***$P < 0.0001$. **(L)** Detection of IFN-β promoter activation in HEK-293T cells was performed 24 hours after transfection with the IFN-β-Luc reporter plasmid, PRV or HSV-1 US1 protein expression plasmids, or an empty vector, with or without co-transfection of cGAS and STING, using the dual-luciferase reporter assay. ns, not significant; ***$P < 0.0001$. **(A-F, H-L)** Representative results from three biological replicates are shown. Data represent mean values ± SD of three technical replicates. Statistical significance was determined by two-tailed unpaired Student's t-test.

US1 inhibits cGAS-STING mediated IFN-β production across different alpha-herpesviruses, we synthetized US1 genes from herpes simplex virus type 2 (HSV-2) and VZV. Remarkably, HSV-2 and VZV US1 also inhibited IFN-β induction in a dose-dependent manner, suggesting a broad inhibitory role of US1 in cGAS-STING mediated IFN-β induction (Fig 1J and 1K). Considering the transcription factor function of US1, to exclude the possibility that US1 itself negatively regulates the activity of the IFN-β promoter, we conducted a dual-luciferase reporter assay in the absence of cGAS and STING transfection. The results showed that, in contrast to the inhibitory effect observed with the addition of cGAS and STING, neither PRV nor HSV-1 US1 alone was able to induce IFN-β promoter activity (Fig 1L). Taken together, these results demonstrate a general role of US1 in suppressing cGAS-STING-mediated IFN-β induction across alpha-herpesviruses.

### US1 interacts directly with cGAS and inhibits its DNA binding and enzymatic activity

To investigate whether US1 interacts with cGAS, we purified recombinant HA-pcGAS-Flag proteins as previously described [32]. Due to the challenges in expressing and purifying HSV-1 US1 in *E.coli* and mammalian cells, we expressed PRV and VZV US1 in *E.coli*, and purified them using gel filtration and ion exchange chromatography (S3A-S3E Fig). Co-immunoprecipitation (Co-IP) demonstrated that PRV US1 directly interacted with pcGAS, hcGAS and mcGAS (Figs 2A, S4A and S4B) and VZV US1 directly interacted with hcGAS (Fig 2B). Additionally, co-transfection of HSV-1 US1 and hcGAS in HEK-293T cells followed by Co-IP showed that HSV-1 US1 also interacted with hcGAS, revealing a broad association between cGAS and US1 across different alpha-herpesviruses (Fig 2C). To determine the specificity of US1 binding to cGAS, we purified CDN-binding domain (CBD) of pSTING (S5A and S5B Fig). No interaction was observed between US1 and pSTING CBD (S5C Fig). Further experiments with PRV US1 co-transfected with full-length pSTING, porcine TBK1 (pTBK1), and porcine IRF3 (pIRF3) in HEK-293T cells showed that PRV US1 did not interact with pSTING, pTBK1, or pIRF3 (S5D-S5F Fig). To provide a physiological context of viral infection, primary porcine alveolar macrophages (PAMs) from specific pathogen-free (SPF) pigs were infected with PRV, and endogenous Co-IP showed US1 interacted with endogenous pcGAS upon PRV infection using anti-US1 pAb (Fig 2D). Furthermore, confocal microscopy of swine testis (ST) cells stably expressing GFP-pcGAS post-PRV infection revealed co-localization of US1 with pcGAS (Fig 2E). Similarly, VZV US1 co-localized with hcGAS in HeLa cells (Fig 2F).

Given US1's direct interaction with cGAS, we next examined its impact on cGAS enzymatic activity. First, we performed enzymatic activity assays using pcGAS, hcGAS and mcGAS incubated with dsDNA, ATP and GTP *in vitro,* followed by the detection of cGAMP via ion exchange chromatography using a Mono-Q column (Figs 2G, S6A and S6B, black peak). PRV US1 protein in the enzymatic reaction of cGAS resulted in a notable reduction in 2'3'-cGAMP production (Figs 2G,

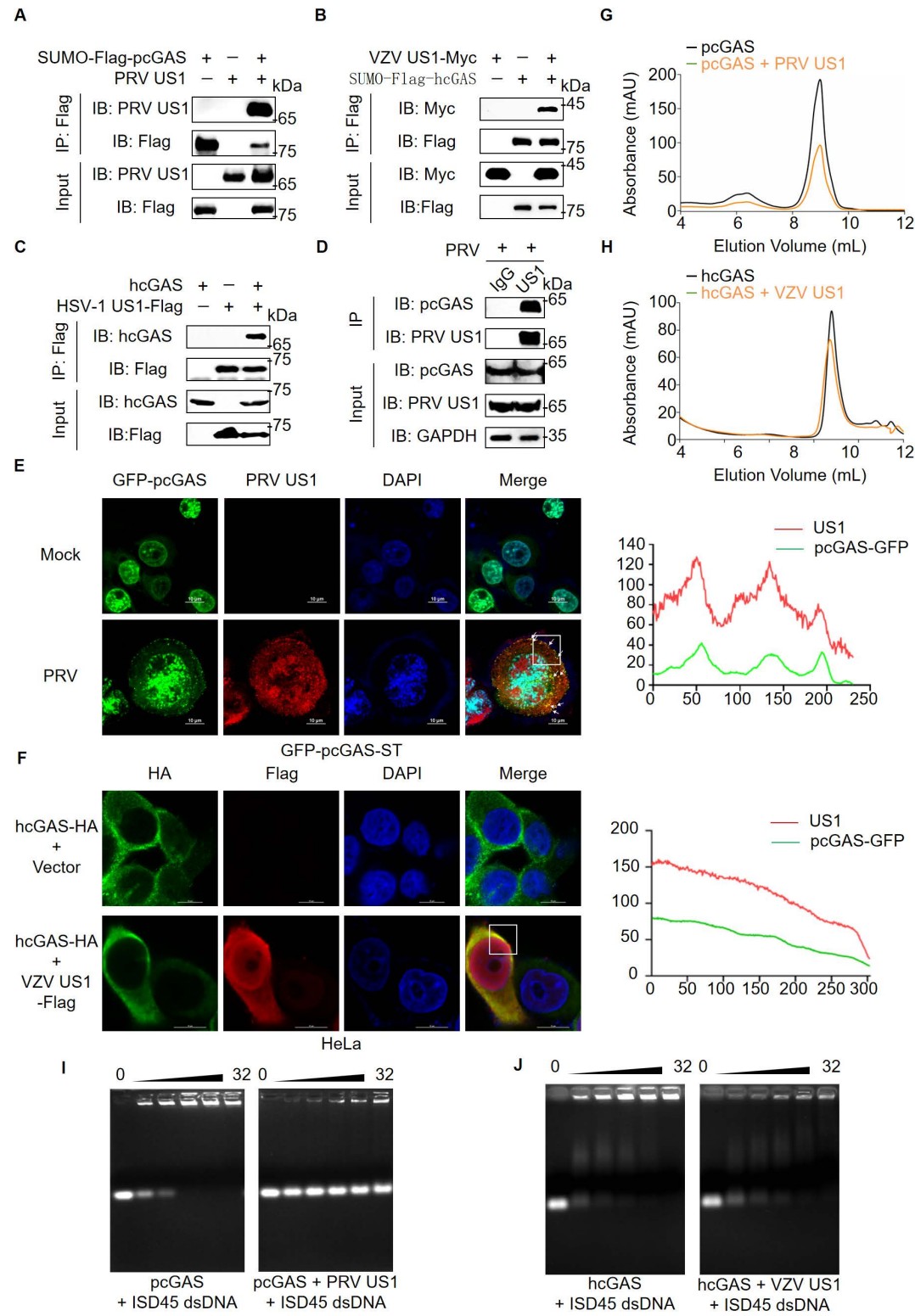

**Fig 2. US1 interacts with cGAS directly and inhibits its DNA binding and enzymatic activity. (A-C)** Co-IP assay for assessment of the interactions between PRV US1 **(A)**, VZV US1 **(B)** or HSV-1 US1 **(C)** and pcGAS or hcGAS. After mixing the purified recombinant proteins or cell lysate in IP buffer

at 4 °C for 2 hours, the mixed buffer was immunoprecipitated using mouse anti-Flag MAb. The immunoprecipitated complex was analyzed by immunoblotting with the indicated antibodies. **(D)** Co-IP assay in the case of PAMs with PRV infection. PAMs were lysed after 8 hours of infection with PRV (MOI = 1), and the supernatants were immunoprecipitated using anti-PRV US1 antibody or anti-IgG control antibody as in **(A)**. **(E)** The confocal microscopy analysis of colocalization between pcGAS and PRV US1. ST cells stably expressing GFP-pcGAS were constructed using lentivirus over-expression system, and then infected with PRV (MOI = 5), followed by detection of co-localization between PRV US1 and pcGAS at 9 h post infection. Nuclei were stained with DAPI (blue). Green fluorescence indicated pcGAS, and red fluorescence indicated PRV US1. The arrows referred to the co-localization of pcGAS and US1. Scale bars, 10 μm. Fluorescence intensity was quantified using ImageJ (right) **(F)** The confocal microscopy analysis of colocalization between hcGAS and VZV US1. HeLa cells were transfected with plasmids encoding hcGAS-HA and an empty vector, or plasmids encoding hcGAS-HA and VZV US1-Flag, respectively. Co-localization between VZV US1 and hcGAS was examined 24 hours post-transfection. The nuclei were stained with DAPI (blue). Green fluorescence indicated hcGAS, and red fluorescence indicated VZV US1. Scale bar, 10 μm. Fluorescence intensity was quantified using ImageJ (right) **(G-H)** cGAS activity assay by ion exchange chromatography. 10 μM pcGAS **(G)** or hcGAS **(H)** was incubated with the Salmon Sperm DNA and equimolar ratio PRV US1 **(G)** or VZV US1 **(H)** proteins in reaction buffer at 37°C for 2 **h.** The reaction product was first purified by ultrafiltration and then analyzed using a MonoQ ion exchange column. **(I-J)** DNA binding analysis of pcGAS **(I)** or hcGAS **(J)** under the influence of PRV US1 **(I)** or VZV US1 **(J)**. In a mixture of 2.5 μM ISD45, US1 and cGAS, proteins were increased in a molar ratio of 1:0 to 1:32, followed by EMSA. **(A-J)** Representative results from three biological replicates are shown.

S6A and S6B, yellow peak). Similarly, VZV US1 also reduced the production of 2′3′-cGAMP by hcGAS (Fig 2H). Next, we conducted electrophoretic mobility shift assays (EMSA) to determine if PRV and VZV US1 affect the DNA binding ability of cGAS. Our results showed that pcGAS, hcGAS and mcGAS bound to interferon stimulated DNA (ISD45) in a dose-dependent manner (Figs 2I, 2J, S6C and S6D, left panels). Notably, co-incubation with PRV US1 significantly reduced the DNA binding ability of cGAS (Figs 2I, S6C and S6D, right panels). Similarly, VZV US1 also reduced the DNA binding of hcGAS (Fig 2J, right panel). Previous studies have shown that both N-terminus (cGAS$^{NT}$) and C-terminus (cGAS$^{CT}$) of cGAS are able to bind to dsDNA [33,34]. We purified pcGAS$^{NT}$ and pcGAS$^{CT}$ protein as previously described (S7A Fig) [32] and demonstrated that both pcGAS$^{NT}$ and pcGAS$^{CT}$ were able to bind dsDNA (S7B and S7C Fig, left panels), whereas PRV US1 inhibited their ability to bind dsDNA (S7B and S7C Fig, right panels). As a negative control, US1 did not bind ISD45 dsDNA (S7D Fig). Similarly, we examined the interaction between recombinant PRV US1 and Flag-tagged full-length pcGAS as well as truncated mutants of pcGAS. Our results showed that while full-length pcGAS effectively bound to PRV US1, the two truncated mutants, pcGAS$^{NT}$ and pcGAS$^{CT}$, exhibited significantly reduced binding compared to full-length pcGAS (S7E Fig). This suggests that both the N-terminal and C-terminal domains of pcGAS are involved in the interaction with PRV US1. Collectively, these results demonstrate that alpha-herpesviruses US1 proteins directly interact with cGAS and inhibit its DNA binding and enzymatic activity.

### Deficiency of peptides in US1 protein increases cGAS-mediated IFN-β production

Despite the low sequence homology between PRV US1 and HSV-1 US1, all variants of US1 across PRV, HSV-1, HSV-2 and VZV inhibited cGAS-mediated IFN-β production. Interestingly, similar to the V protein of Paramyxovirus, which directly binds and disrupts the fold of Melanoma Differentiation-Associated gene 5 (MDA-5), we hypothesized that US1 might inhibit cGAS through a conserved structural feature rather than sequence homology. To test this hypothesis, we used Alphafold (https://github.com/google-deepmind/alphafold [35]) to generate the protein structures of PRV US1 (blue, pLDDT score = 59.9) and HSV-1 US1 (gray, pLDDT score = 50.1), respectively (Fig 3A, left). Surprisingly, two discontinuous peptides within each US1, named peptide A (5 residues) and peptide B (10 residues), overlapped when aligning the two protein structures (Fig 3A, middle and right). We then constructed multiple deletion mutants of US1, including PRV US1 with peptide A deletion (PRV US1$^{ΔA}$), peptide B deletion (PRV US1$^{ΔB}$), and combined peptide A and B deletion (PRV US1$^{ΔAB}$), as well as HSV-1 US1 with corresponding deletions (HSV-1 US1$^{ΔA}$, HSV-1 US1$^{ΔB}$, and HSV-1 US1$^{ΔAB}$ (Fig 3B). As a negative control, we constructed an HSV-1 US1 mutant deleting a peptide corresponding to the same sequence location as peptide A in PRV US1, named HSV-1 US1$^{ΔC}$ (Fig 3B). Our results showed that while PRV US1 significantly reduced pcGAS-mediated IFN-β reporter activation, the three PRV US1 deletion mutants, including PRV US1$^{ΔA}$, PRV

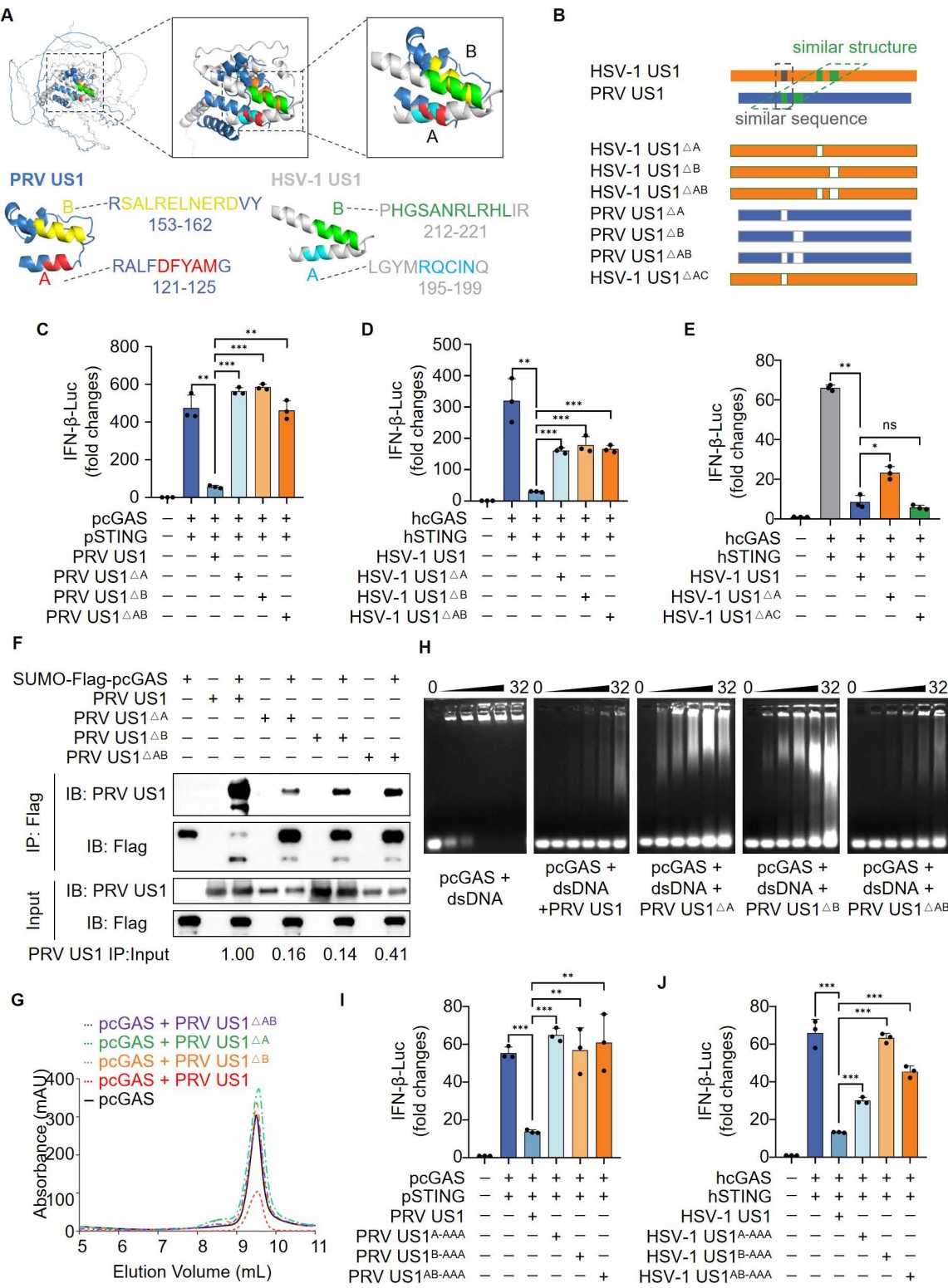

**Fig 3. Deletion of peptides in US1 increases cGAS-mediated IFN-β production. (A)** The structural model of PRV (blue) and HSV-1 US1 (gray) predicted by AlphaFold2. The overlapping regions, designated as peptide A and peptide B, were highlighted in different colors. **(B)** Construction of US1 partial deletion mutants. **(C)** Detection of IFN-β promoter activation in HEK-293T cells transfected with an IFN-β-Luc reporter plasmid, plus expression plasmids for pcGAS, pSTING and PRV US1 or its partial deletion mutants for 24 h by dual-luciferase report assay. Results are presented relative to

co-transfected renilla luciferase expressions. Control vs US1 ($P=0.0005$), US1 vs US1$^{\triangle A}$ ($P<0.0001$), US1 vs US1$^{\triangle B}$ ($P<0.0001$), US1 vs US1$^{\triangle AB}$ ($P=0.0002$). **(D)** HEK-293T cells were treated as in **(C)**, but with HSV-1 US1 replacing PRV US1, and hcGAS/hSTING substituted for pcGAS/pSTING. Control vs US1 ($P=0.0020$), US1 vs US1$^{\triangle A}$ ($P<0.0001$), US1 vs US1$^{\triangle B}$ ($P=0.0007$), US1 vs US1$^{\triangle AB}$ ($P<0.0001$). **(E)** Detection of IFN-β promoter activation in HEK-293T cells transfected with an IFN-β-Luc reporter plasmid, plus expression plasmids for hcGAS, hSTING, HSV-1 US1 or its similar structure deletion mutant (forth column) or similar sequence deletion mutant (fifth column) for 24 h by dual-luciferase report assay. Control vs US1 ($P<0.0001$), US1 vs US1$^{\triangle A}$ ($P=0.0055$), US1 vs US1$^{\triangle AC}$ (ns, not significant). **(F)** Co-IP assay for assessment of the interactions between recombinant pcGAS and PRV US1 or PRV US1 partial deletion mutant proteins. Recombinant proteins were pre-incubated at 4°C in IP lysis buffer for 2 hours followed by Co-IP assay. Densitometric quantitation of PRV US1 were normalized relative to the levels of input by ImageJ. **(G)** cGAS activity assay by ion exchange chromatography. After adding PRV US1 or partial deletion mutant proteins in the enzymatic reaction system, the enzyme activity of pcGAS was determined using a MonoQ ion exchange column. **(H)** DNA binding analysis of pcGAS. In a reaction system containing ISD45 and pcGAS, PRV US1 or partial deletion mutant recombinant proteins with a molar ratio ranging from 1:0 to 1:32 were added, followed by EMSA. **(I, J)** HEK-293T cells were treated as shown in **(C)** or **(D)**, but with the PRV US1 alanine mutant replacing the PRV US1 deletion mutant **(I)** or the HSV-1 US1 alanine mutant replacing the HSV-1 deletion mutant **(J)**. Control vs PRV US1 ($P<0.0001$), PRV US1 vs PRV US1$^{A-AAA}$ ($P<0.0001$), PRV US1 vs PRV US1$^{B-AAA}$ ($P=0.0034$), PRV US1 vs PRV US1$^{AB-AAA}$ ($P=0.0055$), control vs HSV-1 US1 ($P=0.0002$), HSV-1 US1 vs HSV-1 US1$^{A-AAA}$ ($P<0.0001$), HSV-1 US1 vs HSV-1 US1$^{B-AAA}$ ($P<0.0001$), HSV-1 US1 vs HSV-1 US1$^{AB-AAA}$ ($P<0.0001$). **(C-J)** Representative results from three biological replicates are shown. **(C-E, I, J)** Data represent mean values ± SD of three technical replicates. Statistical significance was determined by two-tailed unpaired Student's t-tests.

US1$^{\triangle B}$ and PRV US1$^{\triangle AB}$, lost their inhibitory effect (Fig 3C). Similarly, HSV-1 US1 substantially inhibited hcGAS-mediated IFN-β reporter activation, but this inhibitory effect was significantly reduced in the HSV-1 US1 deletion mutants, including HSV-1 US1$^{\triangle A}$, HSV-1 US1$^{\triangle B}$ and HAV-1 US1$^{\triangle AB}$, compared to WT (Fig 3D). However, the negative control, HSV-1 US1$^{\triangle AC}$ retained its ability to inhibit cGAS-STING-mediated IFN-β reporter activation, similar to HSV-1 US1 WT (Fig 3E). To further confirm the role of peptide deletions in PRV US1 on cGAS function, we expressed and purified three PRV US1 mutants, including PRV US1$^{\triangle A}$, PRV US1$^{\triangle B}$ and PRV US1$^{\triangle AB}$ (S8A-S8F Fig). Co-IP assays showed that deletions of peptides A, B or AB significantly reduced the interaction between PRV US1 and pcGAS (Fig 3F). Additionally, we assessed the effects of PRV US1 deletion mutants on pcGAS enzymatic activity. We observed that all three mutations, PRV US1$^{\triangle A}$, PRV US1$^{\triangle B}$ and PRV US1$^{\triangle AB}$, abolished the ability to inhibit cGAS enzymatic activity, showing levels of enzymatic activity similar to the positive control group (Fig 3G), indicating that these deletions effectively abolished the inhibitory impact on cGAMP synthesis. Consistent with these findings, PRV US1 protein significantly reduced pcGAS binding to ISD45 DNA, whereas the inhibitory effect was significantly diminished in the three deletion mutants (Fig 3H). To minimize the potential structural changes caused by peptide deletion, we constructed eukaryotic expression plasmids in which peptide A, peptide B, or both were mutated to alanine residues. Dual-luciferase reporter assays demonstrated that US1 mutants with alanine substitutions in these regions, despite the absence of peptide deletion, still significantly restored their inhibitory function (Fig 3I and 3J). Taken together, these findings demonstrate that the conserved structural features of US1 are crucial for the interaction with cGAS. The deletion of specific peptides in PRV and HSV-1 US1 significantly reduces their ability to inhibit cGAS activation, leading to increased IFN-β production.

## Peptides derived from US1 bind to cGAS and inhibit its activation

To explore whether viral peptides from PRV US1 directly inhibit cGAS activation, we synthesized three peptides, including peptide A, peptide B and peptide AB from PRV US1, using a random peptide (AWKLQT) as a negative control (Fig 4A). We observed that neither peptide A nor peptide B, nor the random peptide inhibited cGAMP synthesis by cGAS (S9A and S9B Fig). While peptide A and peptide B were solubilized in DMSO, and the random peptide in water, peptide AB was not soluble in both. We resolved this by dissolving peptide AB in 3% ammonia. In contrast to the control where 3% ammonia did not affect cGAS enzymatic activity, peptide AB markedly reduced the enzymatic activity of pcGAS, hcGAS and mcGAS, demonstrating its potent inhibitory effect on cGAS (Figs 4B, 4C, and S9C). To understand why peptide AB inhibits the activity of cGAS, we investigated its impact on cGAS DNA binding using EMSA. Consistently, peptide AB inhibited dsDNA binding by pcGAS, hcGAS and mcGAS, demonstrating a broad inhibitory role across species (Figs 4D, 4E, and

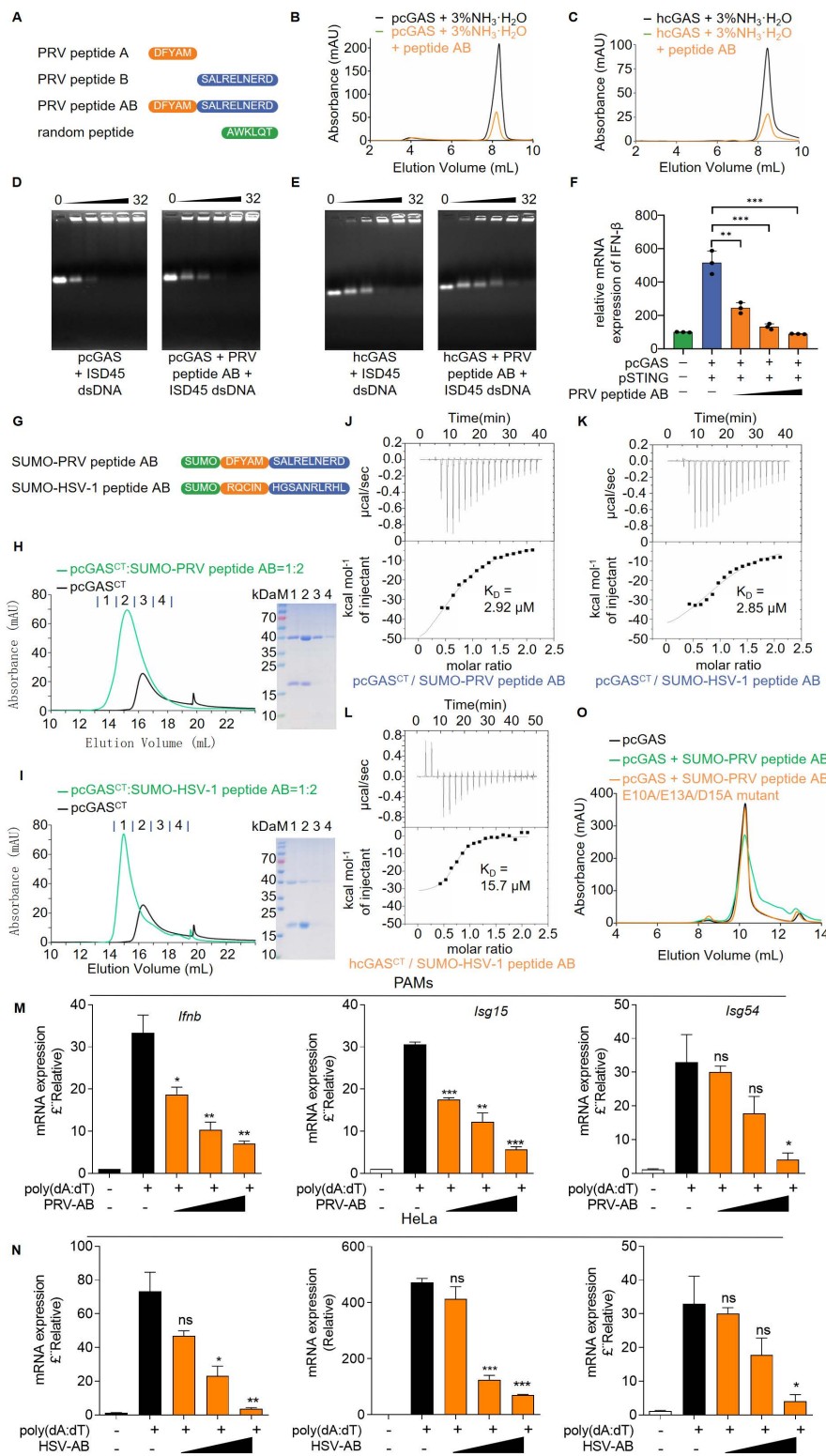

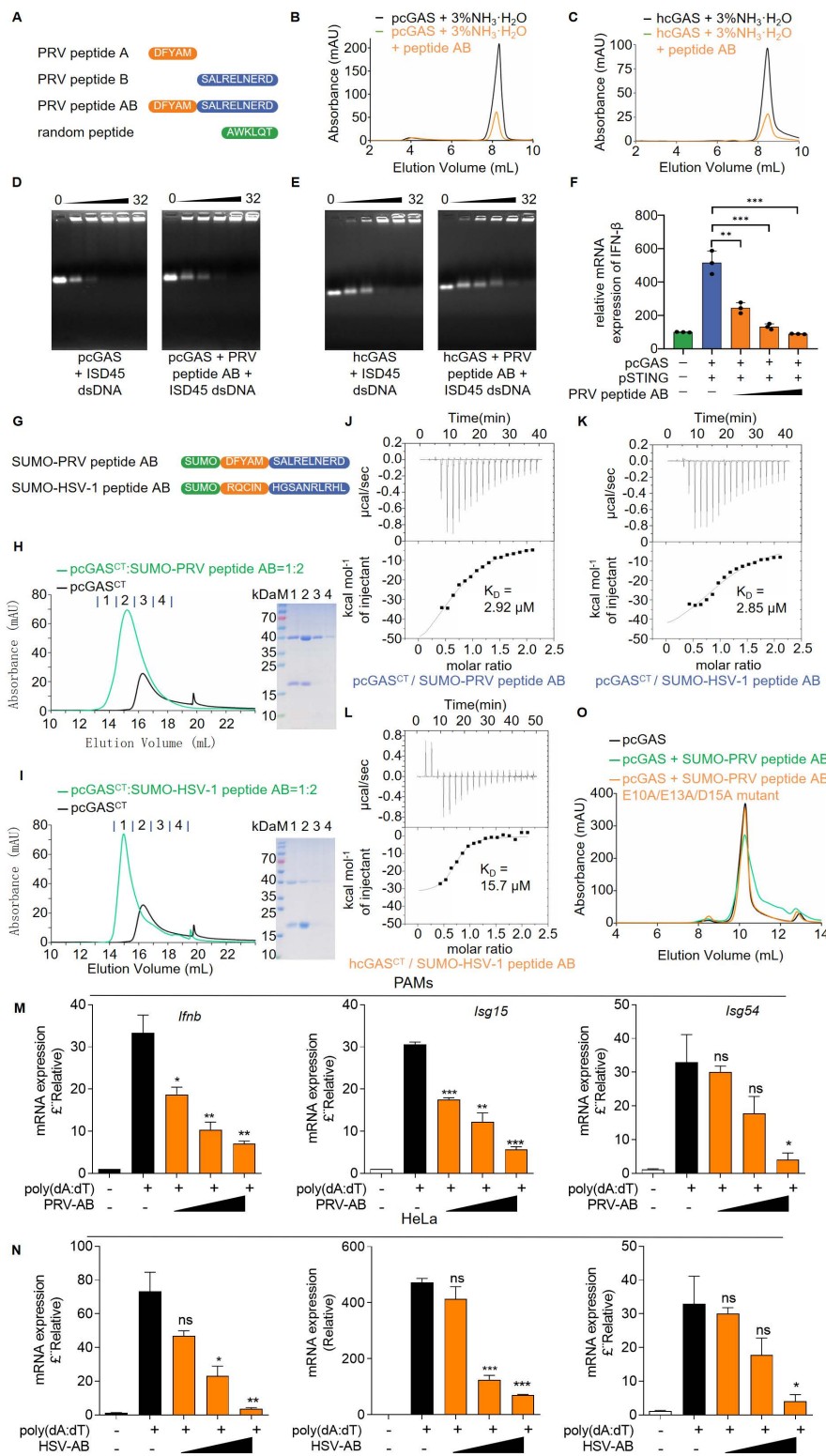

**Fig 4. Peptides derived from US1 bind to cGAS and inhibit its activation. (A)** Schematic representation of peptide design and synthesis. (**B, C**) cGAS activity assay using ion exchange chromatography. PRV peptide AB was dissolved in 3% ammonia water. 10 µM pcGAS (**B**) or hcGAS (**C**) was incubated with the Salmon Sperm DNA and equimolar ratio peptide AB in reaction buffer at 37°C for 2 **h.** The reaction product was first purified by ultrafiltration and then analyzed using a MonoQ ion exchange column. (**D, E**) DNA binding analysis of pcGAS (**D**) or hcGAS (**E**) under the influence of

peptides. In a mixture of 2.5 µM ISD45, peptide AB and cGAS, proteins and peptides were increased in a molar ratio of 1:0 to 1:32, followed by EMSA. **(F)** HEK-293T cells were transfected with increasing doses of peptide AB and expression plasmids encoding pcGAS and pSTING. Cells were harvested after 24 hours for RNA extraction and RT-PCR detection of *IFNB* and *GAPDH* mRNA levels. Data represent mean values ± SD of three technical replicates. Statistical significance was determined by two-tailed unpaired Student's t-tests. Control vs low-peptide AB ($P$ = 0.0036), US1 vs medium-peptide AB ($P$ = 0.0008), US1 vs high-peptide AB ($P$ = 0.0004). **(G)** Illustration of the design for SUMO-peptides. **(H, I)** Gel chromatography analysis to determine whether pcGAS$^{CT}$ binds to SUMO-PRV peptide AB **(H)** or HSV-1 peptide AB **(I)**. pcGAS$^{CT}$ was mixed with SUMO-PRV peptide AB or SUMO-HSV-1 peptide AB at a molar ratio of 1:2 and incubated on ice for 30 minutes. The samples were then subjected to gel chromatography analysis. Blue lines above the gel filtration chromatogram denote the fraction collection ranges for electrophoresis. The corresponding SDS-PAGE lanes (right) are labeled 1-4, corresponding to the first through fourth fractions indicated by blue lines. The shifted chromatographic peak with increased absorbance aligns with the presence of the proteins in the same fractions, confirming the formation of a protein complex. **(J, K)** The binding of cGAS to SUMO-PRV peptide AB **(J)** or SUMO-HSV-1 peptide AB **(K)** was determined using ITC. 100 µM pcGAS$^{CT}$ or 200 µM hcGAS$^{CT}$ was injected 19 times (2 µl per injection) into 300 µl of 10 µM SUMO-PRV peptide AB or SUMO-HSV-1 peptide AB solution. Data were collected at 25°C using a MicroCal ITC 200 titration calorimeter. **(L)** The binding of hcGAS$^{CT}$ to SUMO-HSV-1 peptide AB was determined using ITC as in **(K)**. **(M, N)** PAMs **(M)** or HeLa cells **(N)** were treated with AB peptides derived from PRV US1 or HSV-1 US1 for 6 hours, followed by stimulation with poly(dA:dT) for 12 hours. Cells were then collected, RNA was extracted, and qPCR was performed to assess gene expression.**(O)** cGAS activity was measured using ion-exchange chromatography. Following the addition of SUMO-PRV peptide AB or its E10A/E13A/D15A mutant protein to the enzymatic reaction system, the enzymatic activity of pcGAS was determined using a MonoQ ion-exchange column. (**B-F, H-L, O**) Representative results from at least three biological replicates are shown.

S9D). To further confirmed the inhibitory effect of peptide AB on cGAS-mediated signaling, we utilized qPCR to assess IFN-β transcript levels in cells transfected with plasmids encoding pcGAS, pSTING, and peptide AB. The results showed that peptide AB significantly reduced cGAS-STING-mediated IFN-β mRNA expression in a dose-dependent manner (Fig 4F). Collectively, these results indicate that the synthetic peptide AB derived from PRV US1 directly inhibits cGAS activation.

To further validate the effect of peptide AB on cGAS, we expressed peptides AB derived from PRV US1 and HSV-1 US1 in *E. coli* (S10A-S10E Fig). To assess the inhibitory effects of these peptides when fused with a tag, we coupled a SUMO tag to the peptides (Fig 4G). Both SUMO-PRV peptide AB and SUMO-HSV-1 peptide AB inhibited the enzymatic activity of hcGAS, with mild inhibition observed for pcGAS and mcGAS (S11A-S11C Fig). Subsequently, we conducted cGAS binding study using the catalytic domain of cGAS. We mixed pcGAS$^{CT}$ protein with SUMO-PRV peptide AB or SUMO-HSV-1 peptide AB at a molar ratio of 1:2–2:1 and performed gel filtration chromatography. The results indicated that both SUMO-PRV peptide AB and SUMO-HSV-1 peptide AB bound to pcGAS$^{CT}$, causing a noticeable shift of the cGAS peak (Figs 4H, 4I, and S12A-S12D). This binding was further confirmed by isothermal titration calorimetry (ITC), which revealed that the dissociation constant ($K_D$) of SUMO-PRV peptide AB with pcGAS$^{CT}$ was 2.92 µM, and the $K_D$ of SUMO-HSV-1 peptide AB with pcGAS$^{CT}$ was 2.85 µM (Fig 4J and 4K). Additionally, the $K_D$ value of SUMO-HSV-1 peptide AB with hcGAS$^{CT}$ was 15.7 µM (Fig 4L). As a negative control, ITC results showed no binding between SUMO protein with pcGAS$^{CT}$ (S12E Fig). To verify the inhibitory effect of peptides in cells, PAMs were pre-treated with the PRV US1-derived AB peptide for 6 h, followed by stimulation with poly(dA:dT) for 12 h. The mRNA levels of *Ifnb*, *Isg15* and *Isg54* were significantly reduced compared to the control group, confirming the inhibition of IFN-I by PRV AB peptide (Fig 4M). This inhibitory effect was further confirmed in the context of HSV-1 US1 AB peptide (Fig 4N). Taken together, these results provide functional evidence that the US1-derived peptide effectively inhibits cGAS-mediated innate immune activation both in recombinant protein and cellular levels, ultimately leading to the suppression of IFN-I production.

Given the significant binding observed between both SUMO-PRV peptide AB and SUMO-HSV-1 peptide AB to pcGAS$^{CT}$, we utilized AlphaFold to predict the structural models of the peptide-pcGAS complexes [35]. For the PRV peptide AB-pcGAS complex, two primary models were generated. In one conformation, PRV peptide AB was positioned near the catalytic site of pcGAS (E200, D202), potentially inhibiting its enzymatic function (S12G Fig). In the alternative conformation, PRV peptide AB, through its negatively charged residues (E10, E13, D15), interacted with pcGAS at the R150 and R192 sites, which may interfere with the DNA-binding site of pcGAS (S12H Fig). These distinct models suggested that PRV peptide AB could interact with pcGAS$^{CT}$ at two different sites, aligning with our experimental findings that PRV

peptide AB binds to cGAS and inhibits both its enzymatic activity and DNA-binding capacity. Additionally, the HSV-1 peptide AB-pcGAS complex model indicated that HSV-1 peptide AB adopted a kinked conformation and fitted into the catalytic site (S12F Fig), suggesting an enzyme inhibitor-like function consistent with our experimental results.

To further validate the key residues in PRV peptide AB involved in the interaction with cGAS, we designed a triple-point mutation (E10/E13/D15) based on the model predictions (Figs 4N and S13A-S13C) and performed cGAS enzymatic activity assays. The results showed that, unlike the significant inhibition observed with wild-type PRV peptide AB, the PRV peptide AB E10A/E13A/D15A mutant completely lost its inhibitory activity (Fig 4O). These findings indicate that the critical residues mediating the interaction between PRV peptide AB and cGAS are E10, E13, and D15, which is consistent with our predictive model. Overall, these findings demonstrate that peptides AB derived from US1 directly bind to cGAS and inhibit its activation.

## US1 deficiency enhances the induction of IFN-β and downstream ISGs

To further elucidate the regulatory role of US1 in suppressing cGAS-STING-mediated IFN-β induction, we generated US1-deficient PRV and HSV-1 (PRV-ΔUS1 and HSV-1-ΔUS1) by CRISPR-Cas9 gene knockout technology and DNA homologous recombination, respectively (Fig 5A and 5B). The resulting PRV-ΔUS1 strain was purified through multiple rounds of plaque purification, while HSV-1ΔUS1 was purified using flow cytometry. DNA sequencing and specific PCR confirmed the successful deletion of the US1 gene from both viral genomes (S14 and S15 Fig). To investigate the impact of US1 deficiency on downstream protein activation, we infected THP-1 cells with different doses of PRV-ΔUS1 or PRV-WT. The results showed a significant increase in the phosphorylation of STING, TBK1 and IRF3 upon PRV-ΔUS1 infection compared to PRV-WT (Fig 5C). Similarly, THP-1 cells infected with varying doses of HSV-1-ΔUS1 exhibited significantly enhanced phosphorylation of STING, TBK1 and IRF3 compared to HSV-1 WT (Fig 5D). These findings indicate that US1 negatively regulates the activation of cGAS-STING-mediated signaling. Next, we used qPCR to assess IFN-β production in PAMs infected with PRV-ΔUS1 or PRV-WT, reflecting the situation in primary cells. The results showed that, with or without poly(dA:dT) stimulation, IFN-β mRNA levels were consistently higher in the PRV-ΔUS1 group compared to the PRV-WT group (Fig 5E and 5F). Similarly, the production of IFN-β and downstream ISGs was assessed in L929 cells and PAM-Tang cells infected with either PRV-ΔUS1 or PRV-WT, followed by stimulation with poly (dA:dT). The results showed that poly (dA:dT) induced significantly higher mRNA levels of IFN-β and various ISGs (*MX2*, *ISG15*, *ISG56*, *OAS1b* and *ISG54*) in the PRV-ΔUS1 group compared to the PRV-WT group in both mouse and porcine cells, suggesting a broad inhibitory effect of US1 on IFN-β induction (Figs 5G-5J and S16A-S16F). In addition, we infected L929 cells with HSV-1-ΔUS1 or HSV-1-WT. The mRNA levels of IFN-β and various ISGs (*MX1*, *ISG15*, *ISG56*, *OAS1a* and *CCL5*) induced by HSV-1-ΔUS1 were significantly higher than those induced by HSV-1-WT (Fig 5K-5P). To further validate that US1 does not suppress RNA-driven IFN-β promoter activity, we stimulated cells with poly(I:C) after infection with either PRV-WT or PRV-ΔUS1 and quantified IFN-β expression using RT-qPCR. Unlike the significant differences observed with DNA stimulation, no detectable differences in IFN-β expression were found between PRV-WT and PRV-ΔUS1 under RNA stimulation, confirming that US1 specifically targets DNA-mediated, but not RNA-mediated, IFN-β activation (S16G and S16H Fig). Taken together, these results indicate that US1 effectively suppresses cGAS-mediated IFN-β induction and downstream ISGs expression.

## US1 enhances the pathogenicity of alpha-herpesvirus and suppresses IFN-β induction in mice

To evaluate the impact of US1 on alpha-herpesvirus pathogenicity *in vivo*, C57BL/6 mice were infected intraperitoneally with either PRV-ΔUS1 or HSV-1-ΔUS1 at doses of $0.5 \times 10^4$ PFU or $1 \times 10^7$ PFU per mouse, respectively. Control groups were infected with equivalent doses of PRV-WT or HSV-1-WT, respectively. The survival of the mice was monitored for 15 days to assess the effects of US1 deletion on viral pathogenicity (Fig 6A). Mice infected with PRV-ΔUS1 or HSV-1-ΔUS1 exhibited improved survival rates and less weight loss compared to those infected with PRV-WT or HSV-1-WT (Fig

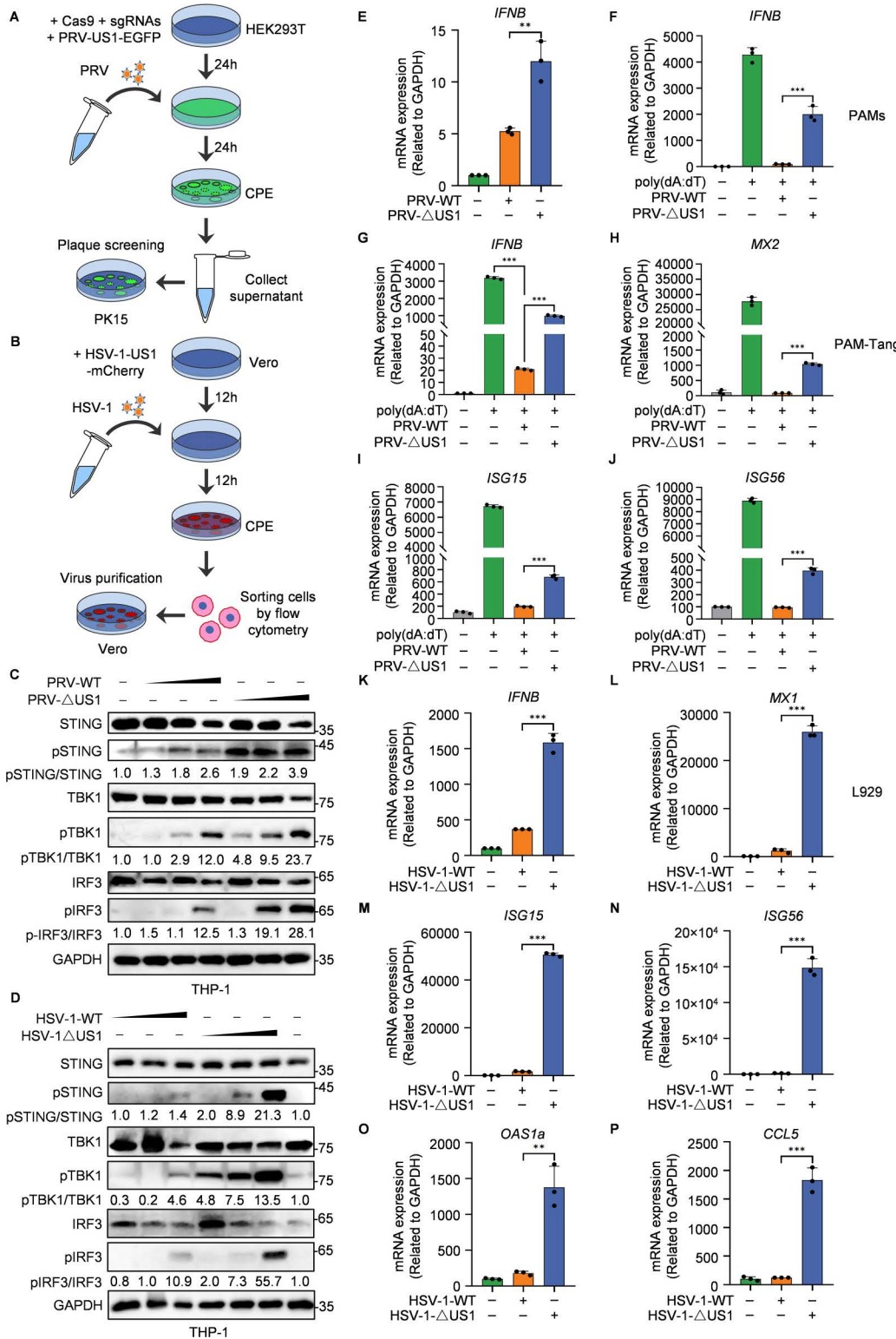

**Fig 5. US1 deficiency enhances the induction of IFN-β and the expression of ISGs. (A)** Construction of a PRV US1 deletion mutant virus using CRISPR-Cas9 system. HEK-293T cells were transfected with sgRNAs and Cas9 co-expression plasmids (pX459-sgRNAs) and US1-homologous-arms-EGFP fusion expression plasmid (PRV-US1-EGFP). PRV (MOI = 1) was infected at 24 hpi, and the supernatant was collected when there was a cytopathic effect (CPE), followed by plaque screening. **(B)** Construction of an HSV-1 US1 deletion mutant virus using homologous recombination

technology. Vero cells were transfected with the US1 homology arm mCherry fusion expression plasmid (HSV-1-US1-mCherry) for 12 hours, followed by infection with HSV-1 (MOI = 1). After 12 hours, when cytopathic effects (CPE) appeared, cells exhibiting mCherry fluorescence were sorted using flow cytometry to purify the virus. **(C, D)** Immunoblot analysis of STING, TBK1, IRF3, and their phosphorylated forms in THP-1 cells infected with either wild-type or US1 deletion mutant PRV **(C)** or HSV-1 **(D)** (MOI = 0.01, 0.1, 1) at 12 hpi. **(E)** PAMs were infected with PRV-WT or PRV-ΔUS1 (MOI = 1) for 12 hours, and the mRNA levels of *IFNB* (relative to GAPDH) were analyzed using RT-qPCR. **\*\*P = 0.0040. (F)** The mRNA levels of *IFNB* (relative to *GAPDH*) in PAMs were analyzed using RT-qPCR. PAMs were first infected with PRV-WT or PRV-ΔUS1 (MOI = 1) for 6 hours, followed by transfection with 1 µg of poly(dA:dT) and incubation for an additional 6 hours. **\*\*\*P = 0.0004. (G-J)** The RT-qPCR analysis of *IFNB* **(G)**, *MX2* **(H)**, *ISG15* **(I)** and *ISG56* **(J)** mRNA expression (related to *GAPDH*) in PAM-Tang cells infected with PRV-WT or PRV-ΔUS1 (MOI = 0.01) for 6h, followed by transfection with 1 µg poly(dA:dT) for an additional 12 hours. **\*\*\*P < 0.0001. (K-P)** The RT-qPCR analysis of *IFNB* **(K)**, *MX1* **(L)**, *ISG15* **(M)**, *ISG56* **(N)**, *OAS1a* **(O)** and *CCL5* **(P)** mRNA expression (related to *GAPDH*) in L929 cells infected with HSV-1-WT or HSV-1-ΔUS1 (MOI = 1) for 12h. **\*\*P = 0.0022; \*\*\*P < 0.0001** (all except CCL5); CCL5: **\*\*\*P = 0.0002. (C-P)** Representative results from three biological replicates are shown. **(E-P)** Data represent mean values ± SD of three technical replicates. Statistical significance was determined by two-tailed unpaired Student's t-tests.

6B-6E). For sample collection, mice were intraperitoneally injected with wild-type virus, deletion virus, or an equivalent volume of DMEM as uninfected controls, using the same doses as in Fig 6A (Fig 6F). Enzyme-linked immunosorbent assay (ELISA) showed that serum IFN-β levels at 24 hours post-infection were higher in mice infected with PRV-ΔUS1 or HSV-1-ΔUS1 compared to those infected with wild-type virus (Fig 6G and 6H). Lung and brain tissues were collected from all euthanized mice 3 or 4 days post-infection for further analysis (Fig 6F). Additionally, mice infected with PRV-ΔUS1 exhibited lower viral loads in both lung and brain tissues compared to those infected with PRV-WT (Fig 6I and 6J). Histo-pathological analysis revealed that PRV-WT infection led to increased inflammatory cells in lung and brain, whereas mice infected with PRV-ΔUS1 displayed milder pathological changes (Fig 6K). Given the neurotropic nature of HSV-1 [36], mice infected with HSV-1-WT exhibited significantly more inflammatory cell infiltration in the brain than those infected with HSV-1-ΔUS1 (Fig 6L). Consistently, mice infected with HSV-1-ΔUS1 had lower viral loads in brain tissue compared to those infected with HSV-1 WT (Fig 6M). While HSV-1 infection did not cause noticeable lung symptoms (S17A Fig), and the viral load in the lungs of mice infected with HSV-1-ΔUS1 was nearly equivalent to that of those infected with HSV-1-WT (S17B Fig). Collectively, these data suggest that US1 enhances the pathogenicity of alpha-herpesvirus and suppresses IFN-β induction in mice.

## Discussion

The cGAS-STING signaling pathway is crucial for the effective control of alpha-herpesvirus infections through the induction of IFN-I [37,38]. However, alpha-herpesviruses have evolved numerous strategies to counteract cGAS-STING signaling. For example, HSV-1 UL37 has been shown to deamidate cGAS, impairing its ability to catalyze cGAMP synthesis in a species-specific manner [13]. PRV UL21 triggers cGAS degradation through toll interacting protein (TOLLIP)-mediated selective autophagy, where cGAS undergoes ubiquitination by the E3 ligase ubiquitin protein ligase E3C (UBE3C) [11]. In this study, we discovered that US1 protein binds directly to cGAS, blocking its DNA binding and enzymatic activities through overlapping peptides in both PRV and HSV-1 US1 structures. These peptides directly antagonize the cGAS-mediated innate signaling pathway, suggesting that this inhibition mechanism by US1 may be a common strategy among the alpha-herpesvirus (Fig 7).

The Zhu group demonstrated in 2015 that KSHV ORF52 is an inhibitor of cGAS (KSHV cGAS inhibitor, KicGAS) that disrupts cGAS-mediated IFN-β induction [19]. Despite the low homology of amino acid sequences within the herpesvirus family, most studies have focused on specific viral proteins rather than on a unified mechanism among herpesviruses, primarily due to the lack of highly homologous motifs. Recent reports suggest that KSHV ORF52 and HSV-1 VP22 prevent cGAS-DNA phase separation, indicating a potential common inhibitory mechanism due to their structural similarity [21], despite ORF52 being conserved only in gamma herpesviruses. However, detailed structural studies on these viral proteins are still lacking. Recent studies have increasingly focused on the structural analysis of viral proteins to elucidate

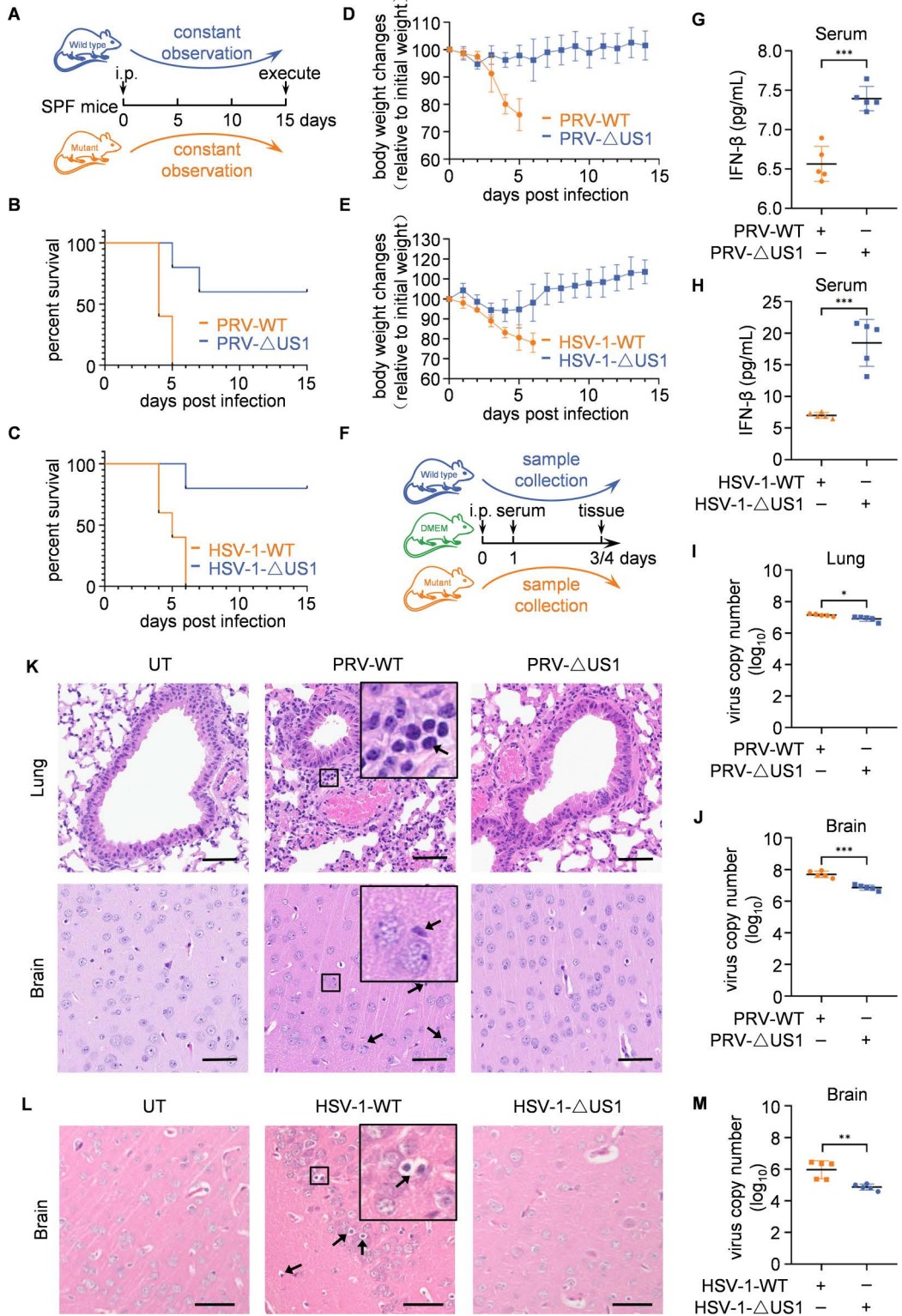

**Fig 6. US1 enhances the pathogenicity of alpha-herpesvirus and suppresses IFN-β induction in mice. (A)** Schematic illustration of the mice survival assay. **(B, C)** Survival rate of C57BL/6 mice ($n = 5$ in each group) infected with PRV-WT and PRV-ΔUS1 (**B**) or HSV-1-WT and HSV-1-ΔUS1 (**C**) at a dose of $0.5 \times 10^4$ plaque forming unit (PFU)/mouse or $1 \times 10^7$ PFU/mouse. **(D, E)** Weight changes of mice over time. Daily weight was normalized as a percentage of the initial weight. The data are from five biological replicates and represent mean values ± SD. Statistical significance was determined

by two-tailed paired Student's t-tests. PRV-WT vs PRV-ΔUS1 ($P=0.0051$), HSV-1-WT vs HSV-1-ΔUS1 ($P=0.0147$). **(F)** Schematic illustration of the sampling process for virus-infected mice. **(G, H)** ELISA of IFN-β in serum from mice infected with PRV-WT and PRV-ΔUS1 **(G)** or HSV-1-WT and HSV-1-ΔUS1 **(H)** at a dose of $0.5\times10^4$ PFU/mouse or $1\times10^7$ PFU/mouse. Serum was collected at 24h after infection. ***$P=0.0001$. **(I, J)** qPCR analysis of viral replication in lung **(I)** and brain **(J)** tissues of mice on the 4th day after infection with PRV-WT or PRV-ΔUS1 at a dose of $0.5\times10^4$ PFU/mouse. *$P=0.0175$, ***$P<0.0001$. **(K)** Hematoxylin and eosin (H&E) staining of brain and lung sections from mice as in **(I)** and **(J)**. Representative images from five biological replicates. Arrows indicated an increase in monocytes in the lungs (top panel) or an increase in glial cells in the brain (bottom panel). Scale bars, 50 μm. **(L)** H&E staining of brain tissues collected 3 days post-infection with either HSV-1-WT or HSV-1-ΔUS1 ($1\times10^7$ PFU/mouse). Representative images from five biological replicates. Arrows indicated an increase in glial cells in the brain. Scale bars, 50 μm. **(M)** qPCR analysis of viral replication in brain tissues from the identical infection groups shown in **(L)**. **$P=0.0032$. **(G-J, M)** Data represent mean values ± SD of five biological replicates. Statistical significance was determined by two-tailed unpaired Student's t-test.

the intricate interactions between viruses and host proteins. However, most studies focus on how viral proteins interact with host proteins, either through analyzing resolved three-dimensional structures or identifying structural domains that are similar to those found in host proteins. Very few studies compare the structures of homologous viral proteins to identify key functional peptides. Previously, it is reported that HSV-2 US1 downregulates IFN-β production by inhibiting the interaction of IRF-3 with the IFN-β promoter, revealing an unconventional mechanism by which HSV-2 evades host innate immunity [39]. HSV-1 US1 can inhibit STING-induced IFN-β activation [40]; however, our study primarily focuses on US1 antagonism of cGAS. In this study, we demonstrated that the US1 proteins from alpha-herpesviruses suppress

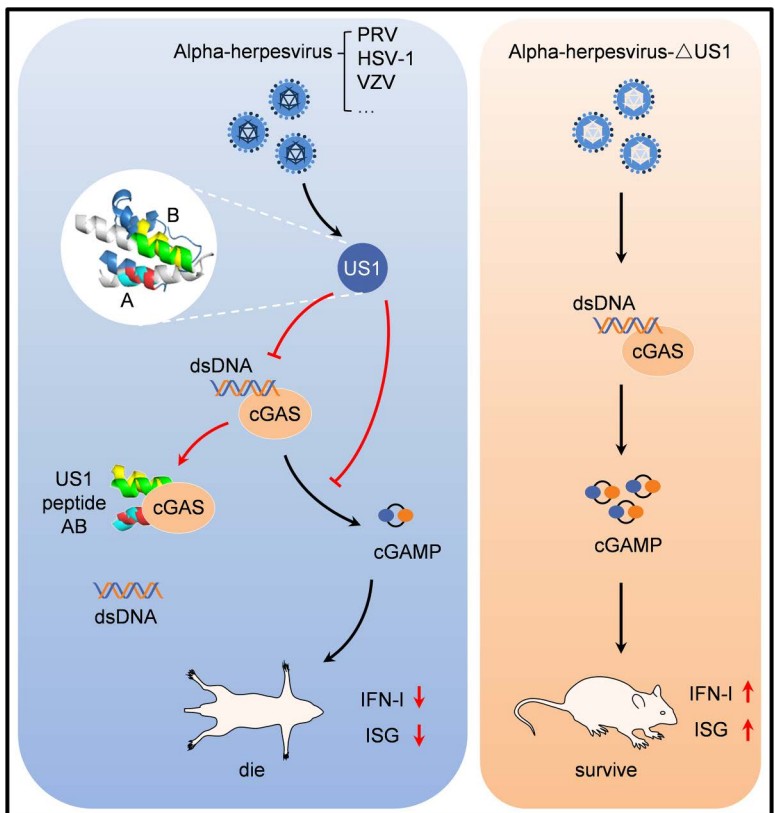

**Fig 7. Proposed working model showing that US1 from alpha-herpesvirus inhibits cGAS-mediated IFN- β induction.**

cGAS activity via two overlapping peptides within their three-dimensional structures. The actual conformation of peptides AB, A, and B remain unknown and may differ from those in the full-length protein. Nevertheless, we have demonstrated the antagonistic effect of these peptides on cGAS through the biochemical and cellular assays. Further research should explore whether similar peptides exist in the US1 proteins of β and γ herpesviruses to determine if a conserved mechanism of cGAS inhibition exists. Our findings suggest a new approach for exploring immune evasion strategies across different herpesvirus subfamilies.

Previously, US1 has been showed to perform diverse roles in virus life cycles. For example, HSV-1 US1 contributes to protein turnover and nuclear remodeling in infected cells [41], and it also plays a companion-like role during viral infection [29]. In addition to influencing the host protein quality control machinery, US1 is required for late viral gene expression [42], modification of RNA polymerase (Pol) II [43,44] and cell cycle regulation [45,46]. These functions highlight US1 as a potential target for antiviral drug development. For the first time, we have demonstrated that US1 inhibits cGAS-mediated IFN-β induction, offering insights into its role in counteracting innate immune responses. As a conserved viral protein in all alpha-herpesviruses, US1 is typically expressed early or immediately after viral entry and plays a crucial role in regulating the transcription and translation of late-stage proteins. Moreover, cGAS is rapidly activated upon viral dsDNA stimulation to initiate immune responses in the early stages of infection. Our study highlights two key biological roles of US1: inhibiting cGAS activation and facilitating viral replication, underscoring its critical function in subverting early innate immune defenses for successful viral infection. While much research has focused on human herpesviruses, studies on animal herpesviruses like PRV, which infects pig, human and mouse, are less extensive. Our findings demonstrate that PRV US1 inhibits pcGAS, hcGAS and mcGAS at cellular, biochemical, and *in vivo* levels, suggesting a broad-spectrum inhibition of cGAS activation.

The potential for developing new vaccines or drugs targeting herpesviruses holds significant economic and public health value. Our findings suggest that the PRV and HSV-1 US1 deletion viruses could be effective candidates for attenuated live vaccines, meriting further investigation. The broad-spectrum inhibition of cGAS by US1 across various species underscores the possibility of developing antiviral treatments targeting alpha-herpesviruses. Moreover, our experiments demonstrated that synthetic peptides derived from PRV or HSV-1 US1 significantly inhibit cGAS activation, highlighting their potential therapeutic potential.

In summary, this study indicates that the US1 proteins of alpha-herpesviruses act as inhibitors of host innate immunity by specially targeting cGAS through overlapping peptides within the US1 structures. This discovery offers valuable insights into the development of anti-herpesvirus drugs and suggests the potential for controlling cGAS-related autoimmune diseases using viral peptides.

## Materials and methods

### Ethics statement

The protocols were approved by the Committee on the Ethics of Animal Experiments of the Harbin Veterinary Research Institute (HVRI) of the Chinese Academy of Agricultural Sciences (CAAS) and the Animal Ethics Committee of Heilongjiang Province, China (Code of Ethics: 231017-01-GR).

### Cells and viruses

Primary PAMs were isolated from 4-week-old specific pathogen-free (SPF) piglets and were cultured in Roswell Park Memorial Institute (RPMI) 1640 (Gibco) containing 20% fetal bovine serum (FBS) (PlantChemMed, #PC-00001) and 1% antibiotics (penicillin and streptomycin) at 37°C with 5% $CO_2$. All cell lines were obtained from the ATCC, authenticated by STR profiling, verified as mycoplasma free and cultured for limited passage numbers. HEK-293T cells (Human embryonic kidney cells), PK-15 cells (Porcine kidney-15), HeLa cells, PAM-Tang cells (Immortalized porcine alveolar macrophages) and L929 cells were cultured in Dulbecco's Modified Eagle Medium (DMEM) (MACGENE, #CM10013) containing 10% FBS and 1% antibiotics at 37°C with 5% $CO_2$. THP-1 cells were cultured in RPMI 1640 containing 15% FBS and 1%

antibiotics at 37°C with 5% $CO_2$. ST cell line stably expressing pcGAS was constructed using lentivirus over-expression system. PRV and HSV-1 were stored in our laboratory. Cell images were analyzed using Falcon S400, Intelligent cell imaging and analysis system (Alicelligent Technologies).

## Plasmids

The C-terminal Hemagglutinin (HA) tagged cGAS, STING, TBK1, IRF3 were ligated to pcDNA3.1 vector at the sites of HindIII and EcoRI in frame. The N-terminal Flag/C-terminal HA double-tagged cGAS and its truncated forms were ligated to His6-SUMO-pET28a vector at the sites of BamHI and HindIII. The C-terminal Flag tagged US1 of PRV, VZV, HSV-2 and HSV-1 were ligated to pCAGGS vector. PRV US1, VZV US1, PRV peptide AB and HSV-1 peptide AB were ligated to His6-SUMO-pET28a vector. Mutant plasmids of PRV and HSV-1 US1 were constructed by overlap PCR.

## Dual-luciferase reporter assays

HEK-293T cells were seeded into 24-well plates. pGL3.0-IFN-β-Luc, pRL-TK, cGAS, STING, and US1 (or its mutants) plasmids were co-transfected using Lipo2000 (Invitrogen, 11668027). At 24 hours after transfection, cells were harvested for luciferase assay. Cell lysates were prepared and analyzed using the Dual-Luciferase Report Assay System (Promega, USA) according to the manufacturer's instructions. The renilla luciferase reporter gene (pRL-TK, Promega) was used as an internal control.

## Generation of US1-deficient virus via CRISPR-Cas9 system and homologous recombination

Two sgRNAs targeting the PRV US1 gene were designed using the gRNA website (https://www.zlab.bio/guide-design-resources). After synthesis (Tsingke Biotech, China), two sgRNAs were respectively ligated into pX459 vector at the sites of BbsI. The upstream and downstream homologous arms of US1 gene were amplified from PRV genomic DNA using PrimeSTAR HS DNA Polymerase (TaKaRa), followed by ligation with EGFP, and then inserted into the pcDNA3.1 vector. HEK-293T cells were transfected with pX459-US1-sgRNA1/2 plasmids and US1-EGFP plasmids for 24 h. Cells were subsequently infected with PRV for 24 h (MOI = 1). The resulting virus mixture was collected and subjected to the plaque assay. Monoclonal viruses expressing green fluorescence were purified by five rounds of fluorescent plaque isolation and identified by PCR and DNA sequencing. The sequences of sgRNAs targeting US1: #1 5'-CACCGAAGCTAAACTCGGACGCGA-3'; #2 5'-CACCGACGGCGAAGAAGACGAAGA-3'. For HSV-1 virus, the upstream and downstream homologous arms of the US1 gene were amplified from the HSV-1 genomic DNA and ligated into the pUC57-mCherry vector. The US1-mCherry plasmid was transfected into Vero cells for 12 hours, after which the cells were infected with HSV-1 for 12 hours (MOI = 1). The virus was purified through five rounds of flow cytometry sorting, followed by identification via PCR and subsequent DNA sequencing.

## Viral infection

Cells were infected with viruses at the indicated multiplicity of infection (MOI). After adsorption for 2 h, the monolayers were overlaid with DMEM supplemental with 1% FBS and incubated at 37°C. For viral titer determination, samples were harvested at the indicated time points [47]. Viruses, released by three cycles of freezing and thawing, were titrated on PK-15 cells or Vero cells by using plaque assay.

## Reverse transcription and quantitative real-time PCR (qPCR)

Total RNA from cells was extracted using the RNAsimple Total RNA Kit (Tiangen Biotech) following the manufacturer's methods. cDNAs were prepared from 1 μg of extracted RNA by using the HiScript II Q RT SuperMix (Vazyme). qPCR for each sample was performed in triplicate with specific primers using Taq Pro Universal SYBR qPCR Master Mix (Vazyme)

on the LightCycler 480 II system (Roche). Relative gene expression was quantified using the $2^{-\Delta\Delta CT}$ method, and GAPDH served as the internal control.

## Protein expression and purification

The sequences encoding pcGAS, hcGAS, mcGAS, VZV US1, PRV US1 and its mutants (ΔA, ΔB, ΔAB) were cloned into His6-SUMO-pET28a vector followed by transformation into *E.coli* BL21(DE3) (Tsingke Biotech, TSC-E01). The bacteria were cultured in LB medium with appropriate Kanamycin. When OD600 reached at 1.0, the proteins were induced overnight at 16°C with 1 mM isopropyl β-D-1-thiogalactopyranoside (IPTG, #18070). The bacteria were lysed in buffer containing 50mM Tris-HCl (pH 8.0), 300 mM NaCl. The proteins were then purified by Ni-NTA beads followed by washing with the buffer containing 20 mM Tris-HCl (pH 7.5), 500 mM NaCl and 25mM imidazole and eluting with the buffer containing 20 mM Tris-HCl (pH 7.5), 150 mM NaCl and 250mM imidazole. The His6-SUMO tag was removed by SUMO protease overnight at 4°C. The processes for SUMO-PRV peptide AB and SUMO-HSV-1 peptide AB were similar to those described above, except that SUMO protease was not used for cleavage. After that, as required, the proteins were further purified by Hiload 16/600 Superdex 200pg gel-filtration chromatography with running buffer containing 20 mM Tris-HCl (pH 7.5), 150 mM NaCl and then purified by resource S ion exchange with buffer A containing 10 mM Tris-HCl (pH 8.0), 100 mM NaCl and buffer B containing 10 mM Tris-HCl (pH 8.0), 1 M NaCl. The purified proteins were concentrated and frozen in liquid nitrogen immediately. All purified proteins were stored in running buffer with 5 mM DTT.

## Co-immunoprecipitation (Co-IP)

HEK-293T cells grown in 6 cm dish were transfected with the indicated plasmids using Lipo2000 (Invitrigen, 11668027), while primary PAMs were infected with PRV. At 24 h post-transfection or 8 h post-infection, cells were collected and lysed with IP lysis buffer (Beyotime, P0013), and the supernatant was collected after centrifugation at 12000 rpm for 10 minutes. The purified recombinant proteins or supernatants were mixed in IP buffer at 4°C for 2 hours. Supernatants were immunoprecipitated with indicated antibodies and protein A/G plus-agarose (Santa Cruz Biotechnology, #sc-2003). After overnight incubation, beads were washed four times with ice-clod PBS. Immunoprecipitates or whole cell lysates were boiled with SDS sample buffer, separated by SDS-PAGE, transferred onto PVDF membranes, and then blotted with specific antibodies.

## Confocal microscopy

ST cells stably expressing GFP-pcGAS (GFP-pcGAS-ST) were seeded into 24-well plates containing slides. When confluence reached up to 40–50%, the cells were infected with PRV (MOI=5). After 9 h, the supernatant was discarded, and the cells were washed with PBS. Alternatively, HeLa cells were co-transfected with HA-tagged hcGAS and Flag-tagged VZV US1 plasmids. After 24 hours, the supernatant was discarded, and the cells were washed. After treatment with 4% fixative solution (Solarbio, #P1110) for 30 min and 0.5% Triton X-100 for 10 min, the cells were washed with PBS and blocked with 1% bovine serum albumin (BSA) (Sigma-Aldrich, #A7906-100G) for 30 min. After completion, the cells were incubated with anti-PRV US1 antibody or corresponding tag antibodies for 2 h and washed three times with PBS, followed by incubation with ALexa Flour 633 goat anti-mouse IgG (H+L) antibody or CoraLite488-conjugated goat anti-rabbit IgG antibody for 30 min and treated with 0.1 µg/ml 4',6-diamidino-2-phenylindole (DAPI) (Beyotime, #C1002) for 5 min. After washing with PBS, the stained cells were observed using a Nikon A1 confocal microscope. Images were collected and analyzed by NIS-Elements AR.

## Electrophoretic mobility shift assays (EMSA)

For the DNA binding assay, 2.5 µM of the 45 bp interferon stimulatory DNA (ISD45), 5'-TACAGATCTACTAGTGATCTA TGACTGATCTGTACATGATCTACA-3' (Synthesized by Sangon Biotech), was mixed with cGAS at molar ratios of 1:0, 1:2, 1:4, 1:8, 1:16, 1:32. The mixtures were resolved on 0.8% agarose gel using an electrophoresis buffer of 40 mM Tris-HCl (pH 9.2) at constant voltage of 110 V. To determine the effect of US1 or peptide AB on dsDNA binding to cGAS, US1 or

peptide AB was added to the mixture of ISD45 and cGAS, and then were resolved using 0.8% agarose gel. The gels were stained with ethidium bromide and documented using GelView 6000Plus.

### The cGAS activity assay

The 10 μM pcGAS, hcGAS and mcGAS were incubated with the Salmon Sperm DNA (Thermo Fisher scientific, #15632011) in reaction buffer containing 20 mM HEPES (pH 7.5), 5 mM $MgCl_2$, 2 mM ATP and 2 mM GTP at 37°C for 2 h. Samples were centrifuged at 12,000 rpm for 10 min. The products in the supernatant were separated by ultrafiltration. The products were further analyzed by MonoQ ion exchange column (GE Healthcare) with running buffer containing 50 mM Tris-HCl (pH 8.5) and followed by elution by gradient NaCl running buffer. The cGAS products (2',3'-cGAMP) were analyzed by ion exchange chromatography.

### Isothermal Titration Calorimetry (ITC)

ITC data were collected using a MicroCal ITC 200 titration microcalorimeter at 25°C. In a typical experiment, a total of 19 injections (each of 2 μl) of pcGAS$^{CT}$ at 100 μM or hcGAS$^{CT}$ at 200 μM were made into a 300 μl solution of SUMO-PRV peptide AB, SUMO-HSV-1 peptide AB, or SUMO at 10 μM. The raw ITC data were processed with Origin 7.0 software (MicroCal), and the curves were fitted to a single-site binding model.

### Analysis of cGAS binding by gel filtration chromatography

cGAS was mixed with SUMO-PRV peptide AB or SUMO-HSV-1 peptide AB at molar ratios of 2:1, 1:1, and 1:2, respectively, and incubated on ice for 30 minutes. The samples were then centrifuged and analyzed using a Superdex 200 Increase 10/300 GL column.

### Animal experiments

All experiments were conducted using 6-week-old C57BL/6 mice, which were maintained in pathogen-free barrier facilities. The mice were randomly grouped. For the survival assay, mice were intraperitoneally infected with PRV-ΔUS1 or HSV-1-ΔUS1 at doses of $0.5 \times 10^4$ PFU or $1 \times 10^7$ PFU per mouse [48,49], respectively. Control groups were infected with the same doses of PRV-WT or HSV-1 WT (n = 5 in each group). The survival time and body weight of mice in each group were recorded daily. For sample collection, mice were intraperitoneally injected with the same doses of wild-type virus, mutant virus, or an equivalent volume of DMEM as an uninfected control (n = 5 in each group). Serum samples were collected at 24 hpi and analyzed for interferon levels by using the Mouse Interferon Beta Serum ELISA Kit (PBL). The lungs and brains from infected mice were collected at 3 or 4 dpi for viral load detection or histopathological examination using hematoxylin and eosin (H&E) staining. The protocols were approved by the Committee on the Ethics of Animal Experiments of the Harbin Veterinary Research Institute (HVRI) of the Chinese Academy of Agricultural Sciences (CAAS) and the Animal Ethics Committee of Heilongjiang Province, China (Code of Ethics: 240529–01-CR).

### Statistical analysis

All the graphs and relevant statistical tests used in the work were created by GraphPad Prism version 8.0.1. Statistical analyses were performed using two-tailed Student's t-tests, with paired or unpaired tests selected based on the experimental design. A P value of < 0.05 was considered to be statistically significant. *P<0.05; **P<0.01 ***P<0.001. ns, not significant.

## Supporting information

**S1 Fig. PRV US1 inhibits cGAS-mediated production of downstream ISGs.** (A, B and C) HEK-293T cells were transfected with plasmids encoding pcGAS, pSTING, and increasing doses of PRV US1. At 24 hours post-transfection (hpt), cells were harvested for RNA extraction, followed by RT-PCR analysis of ISG15 (A), ISG54 (B), ISG56 (C) and GAPDH.

Representative results from three biological replicates are shown. Data represent mean values ± SD of three technical replicates. Statistical significance was determined by two-tailed unpaired Student's t-test. ***P < 0.0001.
(TIF)

**S2 Fig. Amino acid sequence alignment between PRV and HSV-1 US1.** Identical residues were highlighted with a red background, while similar residues were highlighted with a yellow background. α-helices were represented as squiggles. β-strands were depicted as arrows, and strict β-turns were indicated by TT letters.
(TIF)

**S3 Fig. Purification of recombinant proteins including PRV US1, VZV US1, and pSTING-CBD.** (A, B) PRV US1 recombinant proteins were purified using gel filtration (A). The SUMO tag of recombinant proteins was removed by SUMO protease treatment overnight at 4°C followed by SDS-PAGE analysis (B). (C-E) VZV US1 recombinant proteins were purified using gel filtration (C) and ion exchange (D). The SUMO tag of recombinant proteins was removed by SUMO protease treatment overnight at 4°C, followed by SDS-PAGE analysis (E).
(TIF)

**S4 Fig. PRV US1 directly interacts with hcGAS and mcGAS.** (A, B) Co-IP assay for assessment of the interactions between PRV US1 and hcGAS (A) or mcGAS (B). After mixing the purified recombinant proteins in IP buffer at 4 °C for 2 hours, the mixed buffer was immunoprecipitated using mouse anti-Flag MAb. The immunoprecipitated complexes were analyzed by immunoblotting with the indicated antibodies. Results are representative of three biological replicates. Representative results from three biological replicates are shown.
(TIF)

**S5 Fig. PRV US1 does not interact with STING, TBK1, or IRF3.** (A, B) Recombinant proteins pSTING-CBD were purified using gel filtration. The SUMO tag of recombinant proteins was removed by SUMO protease treatment overnight at 4°C. (C) Co-IP assay for the purified recombinant PRV US1 protein and pSTING-CBD protein. After mixing the purified recombinant proteins in IP buffer at 4 °C for 2 hours, the mixed buffer was immunoprecipitated using mouse anti-Flag MAb. The immunoprecipitated complexes were analyzed by immunoblotting with the indicated antibodies. (D-F) Co-IP assay for assessment of the interactions between PRV US1 and pSTING (D), pTBK1 (E) or pIRF3 (F) in HEK-293T cells. After 24 h post-transfection of plasmids expressing two target proteins, the cells were lysed, and the supernatants were treated like the mixed buffer in (C). (C-F) Representative results from three biological replicates are shown.
(TIF)

**S6 Fig. PRV US1 inhibits enzymatic activity and DNA binding of hcGAS and mcGAS.** (A, B) cGAS activity assay using ion exchange chromatography. 10 μM hcGAS (A) or mcGAS (B) was incubated with the Salmon Sperm DNA and equimolar ratio PRV US1 proteins in reaction buffer at 37°C for 2 h. The reaction product was first purified by ultrafiltration and then analyzed using a MonoQ ion exchange column. (C, D) DNA binding analysis of hcGAS (E) or mcGAS (F) under the influence of PRV US1. In a mixture of 2.5 μM ISD45, US1 and cGAS, proteins were increased in a molar ratio of 1:0–1:32, followed by EMSA. (A-D) Representative results from three biological replicates are shown.
(TIF)

**S7 Fig. PRV US1 inhibits pcGAS<sup>NT</sup> and pcGAS<sup>CT</sup> binding to dsDNA and weakly interacts with truncated domains.** (A) Schematic diagram of pcGAS truncated domains. (B, C) DNA binding analysis of pcGAS$^{NT}$ or pcGAS$^{CT}$ under the influence of PRV US1. In a mixture of 2.5 μM 45 bp interferon stimulated DNA (ISD45), PRV US1 and pcGAS$^{NT}$ (B) or pcGAS$^{CT}$ (C), proteins were increased in a molar ratio of 1:0–1:32, followed by EMSA. (D) DNA binding analysis of PRV US1. The reaction system containing ISD45 and PRV US1 recombinant proteins with a molar ratio of 0–16 was treated as in (B). (E) Co-IP analysis to evaluate the interaction between PRV US1 and pcGAS or its truncation mutants. Purified

recombinant proteins were mixed in IP buffer at 4°C for 2 hours, followed by immunoprecipitation of the mixture using mouse anti-Flag MAb. The immunoprecipitated complexes were analyzed by immunoblotting with the specified antibodies. Densitometric quantitation of PRV US1 were normalized relative to the levels of input by ImageJ. (B-E) Representative results from three biological replicates are shown.
(TIF)

**S8 Fig. Purification of peptide deletion mutants of PRV US1 proteins.** (A-F) The SUMO tag of recombinant SUMO-PRV US1 or its deletion mutant proteins was removed by SUMO protease treatment overnight at 4°C. The recombinant proteins PRV US1$^{\Delta A}$ (A), PRV US1$^{\Delta B}$ (C) and PRV US1$^{\Delta AB}$ (E) were purified using gel filtration, followed by SDS-PAGE analysis (B, D and F).
(TIF)

**S9 Fig. PRV peptide AB inhibits enzymatic activity and DNA binding of mcGAS.** (A, B) cGAS activity assay by ion exchange chromatography. 10 μM pcGAS was incubated with the Salmon Sperm DNA and equimolar ratio peptide A, pep-tide B (A) or random peptide (B) in reaction buffer at 37°C for 2 h. The reaction product was first purified by ultrafiltration and then analyzed using a MonoQ ion exchange column. (C) Peptide AB was dissolved in 3% ammonia water, and then added to the enzymatic reaction system for mcGAS enzyme activity assay by ion exchange chromatography. cGAS activity assay was conducted as described in A, with an equal amount of ammonia water added as a control. (D) DNA binding analysis of mcGAS under the influence of PRV peptide AB. In a mixture of 2.5 μM ISD45, peptide and cGAS, peptide weas increased in a molar ratio of 1:0–1:32, followed by EMSA. (A-D) Representative results from three biological replicates are shown.
(TIF)

**S10 Fig. Purification of SUMO-tagged peptides.** (A-D) The recombinant proteins SUMO-PRV US1 peptide AB (A) and SUMO-HSV-1 US1 peptide AB (C) were purified using gel filtration, followed by SDS-PAGE analysis (B and D). (E) Control SUMO proteins were obtained by overnight cleavage at 4°C using SUMO protease.
(TIF)

**S11 Fig. SUMO-PRV peptide AB and SUMO-HSV-1 peptide AB inhibit enzymatic activity of cGAS.** (A-C) cGAS activity assay using ion exchange chromatography. 10 μM hcGAS (A), pcGAS (B) or mcGAS (C) was incubated with the Salmon Sperm DNA and equimolar ratio SUMO-PRV peptide AB or SUMO-HSV-1 peptide AB in reaction buffer at 37°C for 2 h. The reaction product was first purified by ultrafiltration and then analyzed using a MonoQ ion exchange column. Results are representative of three biological replicates. Representative results from three biological replicates are shown. Representative results from three biological replicates are shown.
(TIF)

**S12 Fig. SUMO-PRV peptide AB and SUMO-HSV-1 peptide AB bind to pcGAS$^{CT}$.** (A-D) Gel filtration chromatography analysis to determine whether pcGAS$^{CT}$ binds to SUMO-PRV peptide AB (A, C) or HSV-1 peptide AB (B, D). pcGAS$^{CT}$ was mixed with SUMO-PRV peptide AB or SUMO-HSV-1 peptide AB at a molar ratio of 2:1 or 1:1 and incubated on ice for 30 minutes. The samples were then analyzed by gel filtration chromatography. Blue lines above the gel filtration chromato-gram denote fraction collection ranges for electrophoresis. The corresponding SDS-PAGE lanes (right) are labeled 1–4, corresponding to the first through fourth fractions indicated by blue lines. The shifted chromatographic peak with increased absorbance aligns with the presence of the proteins in the same fractions, confirming the formation of a protein complex. (E) The interaction between pcGAS and SUMO was analyzed using ITC. A 100 μM solution of pcGAS$^{CT}$ was titrated into 300 μl of 10 μM SUMO solution, with 2 μl injected per titration. Data were collected at 25°C using a MicroCal ITC 200 isothermal titration calorimeter. (F) The complex structural model of HSV-1 peptide AB-pcGAS predicted by AlphaFold2. Yellow indicated HSV-1 peptide AB, gray indicated pcGAS$^{CT}$, and red indicated the enzyme active sites of pcGAS. (A-E) Representative results from three biological replicates are shown. (**G, H**) The complex structural models of PRV peptide

AB-pcGAS predicted by AlphaFold2. Blue indicated PRV peptide AB, gray indicated pcGAS$^{CT}$, red indicated enzyme active sites of pcGAS, and pink indicated the DNA binding sites of pcGAS.
(TIF)

**S13 Fig. Purification of SUMO-PRV peptide AB E10A/E13A/D15A mutant.** (A) Nickel column purification of SUMO-PRV peptide AB E10A/E13A/D15A mutant proteins. (B, C) The recombinant proteins SUMO-PRV US1 peptide AB E10A/E13A/D15A mutant were purified using gel filtration (B), followed by SDS-PAGE analysis (C).
(TIF)

**S14 Fig. Identification of PRV-ΔUS1 deletion mutant virus.** (A) Schematic illustration of strategy used to construct the PRV-ΔUS1 deletion mutant virus. (B) Monoclonal viruses expressing green fluorescence were isolated using fluorescence microscopy. (C) Verification of the US1 deletion in the mutant PRV genome was performed by PCR assay with primers designed based on the full-length US1 sequence, followed by DNA sequencing.
(TIF)

**S15 Fig. Identification of HSV-1-ΔUS1 deletion mutant virus.** (A) Schematic illustration of the strategy used to construct HSV-1-ΔUS1 deletion mutant virus. (B) Monoclonal viruses expressing red fluorescence were identified. (C) Verification of the US1 deletion in the mutant HSV-1 genome was performed by PCR assay using primers designed based on the full-length US1 sequence.
(TIF)

**S16 Fig. PRV US1 deficiency enhances IFN-β induction and downstream ISGs in PAM-Tang and L929 cells.** (A, B) The RT-qPCR analysis of *OAS1b* (A) and *ISG54* (B) mRNA expression (related to *GAPDH*) in PAM-Tang cells infected with PRV-WT or PRV-ΔUS1 (MOI = 0.01) for 6h, followed by transfection with 1 μg poly(dA:dT) for an additional 12 hours. (C-F) The RT-qPCR analysis of *IFNB* (C), *MX1* (D), *OAS1a* (E) and *CCL5* (F) mRNA expression (related to *GAPDH*) in L929 cells infected with PRV-WT or PRV-ΔUS1 (MOI = 0.01) for 6h, followed by transfection with 1 μg poly(dA:dT) for 12 hours. (G, H) PAM-Tang or L929 cells were infected with either PRV-WT or PRV-ΔUS1 (MOI = 0.01) for 6 hours, followed by transfection with 1 μg poly(I:C) for 12 hours, the mRNA expression level of *IFNB* was quantified by RT-qPCR and normalized to *GAPDH*. (A-H) Representative results from three biological replicates are shown. Data represent mean values ± SD of three technical replicates. Statistical significance was determined by two-tailed unpaired Student's t-tests. ***$P < 0.0001$, **$P = 0.0089$ (D), **$P = 0.0041$ (E), **$P = 0.0010$ (F). ns, not significant.
(TIF)

**S17 Fig. Viral propagation in lungs of HSV-1-ΔUS1-infected mice is lower than in HSV-1-WT-infected mice.** (A) H&E staining of lung tissues collected 3 days post-infection with either HSV-1-WT or HSV-1-ΔUS1 ($1 \times 10^7$ PFU/mouse). Representative images from five biological replicates. Scale bars, 50 μm. (B) qPCR analysis of viral replication in lung tissues from the identical infection groups shown in (A). Data represent mean values ± SD of five biological replicates.
(TIF)

## Acknowledgments

The authors are grateful to Professor Yandong Tang from the Harbin Veterinary Research Institute (China) for generously providing the PAM-Tang cells [50].

## Author contributions

**Conceptualization:** Xin Li.

**Data curation:** Weiyu Qu, Ye Yuan, Dongdong Shen, Jiwen Zhang, Xiangyu Huang.

**Formal analysis:** Weiyu Qu, Ye Yuan, Jiwen Zhang, Xiangyu Huang.

**Funding acquisition:** Xin Li.

**Investigation:** Weiyu Qu, Ye Yuan, Dongdong Shen, Jiwen Zhang, Xiangyu Huang.

**Methodology:** Lei Wu, Hongyan Yin, Zhenchao Zhao, Haiwei Wang, Lvye Chai, Jiayang Wu, Xijun He, Cheng Zhu.

**Software:** Weiyu Qu, Ye Yuan, Xiangyu Huang.

**Supervision:** Dongming Zhao, Xin Li.

**Validation:** Weiyu Qu, Ye Yuan, Xiangyu Huang.

**Visualization:** Weiyu Qu, Ye Yuan, Xiangyu Huang.

**Writing – original draft:** Weiyu Qu, Xiangyu Huang, Xin Li.

**Writing – review & editing:** Weiyu Qu, Dongming Zhao, Xin Li.

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
