## [Decision Letter · Decision Letter 0]

15 Jul 2025

Alpha-herpesvirus US1 interacts with cGAS to suppress type I IFN responses and antiviral defense

PLOS Pathogens

Dear Dr. Li,

Thank you for submitting your manuscript to PLOS Pathogens. After careful consideration, we feel that it has merit but does not fully meet PLOS Pathogens's publication criteria as it currently stands. Therefore, we invite you to submit a revised version of the manuscript that addresses the points raised during the review process.

Please submit your revised manuscript within 60 days Sep 13 2025 11:59PM. If you will need more time than this to complete your revisions, please reply to this message or contact the journal office at plospathogens@plos.org. Please include the following items when submitting your revised manuscript:

We look forward to receiving your revised manuscript.

Kind regards,

Pinghui Feng

Academic Editor

PLOS Pathogens

Alison McBride

Section Editor

PLOS Pathogens

Editor-in-Chief

PLOS Pathogens

orcid.org/0000-0003-2946-9497

Editor-in-Chief

PLOS Pathogens

orcid.org/0000-0002-7699-2064

**Journal Requirements:**

At this stage, the following Authors/Authors require contributions: Jiwen Zhang, Lei Wu, Zhenchao Zhao, Haiwei Wang, Lvye Chai, Jiayang Wu, Xijun He, Cheng Zhu, Dongming Zhao, and Xin Li. Please ensure that the full contributions of each author are acknowledged in the "Add/Edit/Remove Authors" section of our submission form.

https://journals.plos.org/plospathogens/s/submission-guidelines#loc-parts-of-a-submission

3) Some material included in your submission may be copyrighted. According to PLOSu2019s copyright policy, authors who use figures or other material (e.g., graphics, clipart, maps) from another author or copyright holder must demonstrate or obtain permission to publish this material under the Creative Commons Attribution 4.0 International (CC BY 4.0) License used by PLOS journals. Please closely review the details of PLOSu2019s copyright requirements here: PLOS Licenses and Copyright. If you need to request permissions from a copyright holder, you may use PLOS's Copyright Content Permission form.

Potential Copyright Issues:

- Figures 5 and 7. Please confirm whether you drew the images / clip-art within the figure panels by hand. If you did not draw the images, please provide (a) a link to the source of the images or icons and their license / terms of use; or (b) written permission from the copyright holder to publish the images or icons under our CC BY 4.0 license. Alternatively, you may replace the images with open source alternatives. See these open source resources you may use to replace images / clip-art:

4) Please ensure that the funders and grant numbers match between the Financial Disclosure field and the Funding Information tab in your submission form. Note that the funders must be provided in the same order in both places as well. State what role the funders took in the study. If the funders had no role in your study, please state: "The funders had no role in study design, data collection and analysis, decision to publish, or preparation of the manuscript.".

**Reviewers' Comments:**

Reviewer's Responses to Questions

**Part I - Summary**

Reviewer #1: n this manuscript, the authors reported that US1 protein from HSV-1 and PRV antagonizes the type I interferon response. US1 interacted with cGAS and inhibited its dsDNA binding and enzymatic activity. The authors further identified a conserved overlapping region in US1 form both PRV and HSV-1 that is necessary and sufficient to inhibit cGAS. For in vivo studies, US1-deficent PRV and HSV-1 induced higher innate immune response in infected mice. While the study offers new insights into innate immune evasion by alpha-herpesviruses, the conclusions would benefit from additional validation.

Reviewer #2: In this manuscript, Qu et al. reported alpha-herpesviruses US1s have shared mechanisms to interact with and block cGAS-mediated antiviral signaling. By alpha-fold prediction, the authors also defined a structural homologous region of US1 and developed corresponding peptides which inhibited cGAS activity across species. Lastly, the authors also produced US1 deficient viruses and showed the necessary roles of US1 on viral replication and immune escape in cell culture and in mice. Overall, the study is very comprehensive with mostly convincing data. On the other hand, there are a few incoherent results or conclusions requiring attention. My detailed comments are listed below:

Reviewer #3: In this manuscript, the authors identified the US1 proteins of PRV and HSV-1 as inhibitors of cGAS. By AlphaFold prediction, they identified structurally conserved regions necessary for cGAS inhibition.

They also generated US1-deleted PRV and HSV-1 mutants and showed a stronger activation of the cGAS-STING pathway in ΔUS1-infected cells. They also showed that ΔUS1 are attenuated in mice.

Strengths: The biochemical assays demonstrating an interaction of the US1 protein with cGAS and inhibition of the cGAS-STING pathway in transfected cells are generally strong and convincing.

Weakness: Although the authors tried to demonstrate the biological importance of US1-mediated cGAS inhibition for viral replication and pathogenesis, the results remain inconclusive. The authors did not consider that PRV and HSV-1 express additional inhibitors of the cGAS-STING signaling pathways, nor did they consider that US1 has additional functions besides inhibiting this pathway. Hence, the reason for the attenuation of the ΔUS1 viruses remains unclear.

As the manuscript contains a large set of data as it stands, the authors could consider splitting the paper into two. One paper could describe the biochemical results on US1 and its interaction with cGAS in transfected cells. This study would be suitable for a biochemical journal. A second paper could then describe the importance of US1-mediated cGAS inhibition on viral replication in vitro and pathogenesis in vivo. This would require several additional experiments (see below) to make the results conclusive and the study appropriate for PLoS Pathogens.

**Part II – Major Issues: Key Experiments Required for Acceptance**

Reviewer #1: 1.The inhibition of IFN-β promoter activity by US1 shown in Figure 1, although statistically significant, lacks validation in different cell lines or experimental conditions. It is recommended to verify the inhibitory effect of US1 on the cGAS-STING signaling pathway in other relevant cell lines.

2.While the authors demonstrate that US1 interacts with cGAS, and have identified a potential region of US1 interacting with cGAS. However, further truncation of cGAS followed by interaction assays would provide more detailed molecular interaction insights.

3.In FIG 4, additional cellular experiments are needed to substantiate the conclusions. For example, innate immune activation with DNA viruses or DNA mimics such as HT-DNA, followed by treatment with US1-derived peptides, would provide more compelling functional evidence.

4.In figure 5, Although the authors have observed a significant increase in IFN-β production upon infection with US1-deficient viruses (PRV-ΔUS1 and HSV-1-ΔUS1), they have not assessed how US1 deficiency affects viral replication. Does US1 deficiency lead to significant attenuation of viral replication? Furthermore, whether the deletion of key innate immune components (e.g., TBK1 or IRF3) at least partially rescue the replication of US1-deficient viruses? Addressing these questions would clarify whether the enhanced immune response is causally linked to US1's immune antagonism.

5.The in vivo experiments in FIG 6 primarily measured IFN-β levels and viral load. However, the results may not specifically reflect the antagonization of cGAS by US1, considering the multifaceted functions of US1. Since the authors have defined the cGAS interacting region of US1 in Figure 3. The authors may consider generating a mutant HSV-1 with selective disruption of this interaction (e.g., point mutations in the binding domain) rather than a complete US1 knockout. Although not perfect, such a mutant virus would provide a more precise tool to evaluate the immunomodulatory role of US1 both in vitro and in vivo..

Reviewer #2: (1) HSV-2 US1 has been reported to block IRF3 for IFN-I inhibition. Another study also shows that HSV-1 US1 blocks STING-induced interferon responses (PMCID: PMC6854482). The authors should discuss these discrepancies. Moreover, for Figure 1 most of the reporter assays were done in both cGAS and STING overexpression manner. The authors should try STING overexpression along with US1 to probe whether US1 also inhibits STING-mediated responses.

(2) (Line 153) It is confusing to me as the author could readily express Flag-tagged HSV-1 US1 but fail to ‘express’ and ‘purify’ the protein. As VZV US1 did not demonstrate optimal anti-cGAS activities compared to PRV US1 (Figures 2H and 2J), purified HSV-1 US1, if possible, could provide a much stronger inhibitory effect.

(3) Figure 2E, it seemed that the overexpressed cGAS and US1 were mutually exclusive in the nucleus, which did not substantially support the authors’ conclusion on colocalization; moreover, Figure 2F was also weak provided the fact that both proteins demonstrated whole cytoplasmic distribution. It is better to overexpress US1 in a cell line with good signals for endogenous cGAS and also to perform quantification of the signal overlapping percentage.

(4) Figure S7E, if PRV US1 had significantly reduced binding to both pcGAS NT and CT, how did it block NT and CT from binding to dsDNA (Figures S7B and S7C)?

(5) Figure 2J is hard to interpret as VZV US1 inhibited the formation of cGAS-dsDNA complex but did not result in more naked dsDNA in the gel.

(6) Figure 3C and 3D need protein expression immunoblots as expression levels of the mutants determine their potency in antagonizing cGAS.

(7) Figure 3G showed a much broader peak for cGAMP which was not observed in any other cGAS activity assay results.

(8) The negatively charged residues ‘EED’ are not present in the proposed overlapping region of HSV-1 US1. Thus, the models predicted by Figures 4M and 4N may not apply to HSV-1 US, which weakens the hypothesis that this aligned region is the conserved structural component inhibiting cGAS.

Reviewer #3: 1. The introduction mentions alphaherpesvirus genes and proteins inhibiting the cGAS-STING signaling pathway without describing their mechanism of action. This information is important for the understanding of the present study. It is important to understand whether the different inhibitors function in a complementary or redundant fashion. It is also necessary to describe other known functions of US1 in the introduction (not only in the discussion).

2. Figure 5. Several PRV and HSV-1 proteins target the cGAS-STING pathway, suggesting that they have redundant functions. Why is the deletion of US1 sufficient to activate the pathway in infected cells? Do the ΔUS1 viruses replicate to lower titers in THP-1 (or other) cells, and can the replication defect be rescued by knocking out of cGAS or STING?

3. Figure 6. The ΔUS1 viruses are attenuated in vivo. However, considering that US1 has additional functions, it is not clear whether the attenuation is caused by a lack of cGAS inhibition. This could be addressed by using a virus expressing a mutant US1 protein that is unable to inhibit cGAS but retains other US1 functions. Alternatively, the authors could demonstrate a (partial) rescue of ΔUS1 viruses in cGAS or STING KO mice.

4. Lines 246-252. It is very confusing that alanine substitution mutants behaved essentially like WT US1. Doesn’t this challenge the importance of the peptides?

**Part III – Minor Issues: Editorial and Data Presentation Modifications**

Reviewer #1: 1.The phrase “overlapping peptides within PRV and HSV-1 US”, in the abstract and elsewhere, is misleading. Since the identified region is part of the US1 protein and not a naturally processed peptide, it would be more accurate to refer to it as a “conserved overlapping region”.

2.Similarly, line 102-103, the phrase “utilizing viral peptides to antagonize cGAS detection” is also inappropriate. This region is not a peptide generated during viral infection, while the use of synthetic peptides in experimental assays is acceptable.

3.The Materials and Methods section should include information on the primers used in the study.

Reviewer #2: (1) Line 144 (Figure 1L), as the basal level of IFN-beta was low in HSV-1-infected cells, it is not strong evidence to claim that US1 fails to suppress IFN-beta in the absence of cGAS/STING overexpression.

(2) Line 244, as all three PRV US1 mutants demonstrated partial inhibition on cGAS to bind dsDNA (Figure 3H), the conclusion that the mutants did not exhibit reduction is inaccurate.

(3) Without a linker sequence between peptide A and B, it would be challenging to adapt to a kinked helix structure as presented in Figure 3A. As such, the contribution of A in the peptide AB may not reflect or mimic what truly happens in the context of the whole protein. The authors should consider this possibility when describing the results.

(4) It seems that peptide AB blocked cGAS binding to dsDNA at a much lower efficiency compared to the full protein (Figure 4D). Nevertheless, it potently inhibited cGAMP production. According to Figure M and N, the peptides could interact with cGAS at two different sites, which may explain their potency in antagonizing cGAS activity. However, in the context of a full protein, it is unlikely that both mechanisms exist considering the steric hindrance of protein-protein interactions. The authors should discuss this point when describing the result.

Reviewer #3: 5. Line 63. “genital infections” rather than “reproductive system infections”.

6. Line 108. Inhibition of the cGAS-STING pathway by alphaherpesvirus proteins has been shown before. Hence, the authors did not uncover a novel strategy to counteract innate immunity.

7. Fig. 1A and B. There are no positive controls.

8. Line 169. Interacted endogenously with pcGAS -> interacted with endogenous pcGAS

9. Line 172-173 and Figs. 2E and F. Colocalization is not convincing. Both proteins are cytoplasmic.

10. Line 347. we challenged L929 cells with HSV-1-ΔUS1 or HSV-1-WT -> we infected

11. Discussion, line 418-425. Delete this part as it is not relevant for the present study.

PLOS authors have the option to publish the peer review history of their article (what does this mean? ). If published, this will include your full peer review and any attached files.

**Do you want your identity to be public for this peer review?** For information about this choice, including consent withdrawal, please see our Privacy Policy .

Reviewer #1: No

Reviewer #2: No

Reviewer #3: No

**Figure resubmission:**

**Reproducibility:**



---

## [Decision Letter · Decision Letter 1]

27 Oct 2025

Dear Dr. Li,

We are pleased to inform you that your manuscript 'Alpha-herpesvirus US1 interacts with cGAS to suppress type I IFN responses and antiviral defense' has been provisionally accepted for publication in PLOS Pathogens. If you can, please address the questions from reviewer #4 via editing your manuscript.

Best regards,

Pinghui Feng

Academic Editor

PLOS Pathogens

Alison McBride

Section Editor

PLOS Pathogens

Sumita Bhaduri-McIntosh

Editor-in-Chief

PLOS Pathogens

orcid.org/0000-0003-2946-9497

Michael Malim

Editor-in-Chief

PLOS Pathogens

orcid.org/0000-0002-7699-2064

Reviewer Comments (if any, and for reference):

Reviewer's Responses to Questions

**Part I - Summary**

Reviewer #1: The authors have addressed my concerns, and I have no additional comments.

Reviewer #2: The authors have adequately addressed all my concerns.

Reviewer #3: The authors responded to all my questions and concerns, and I am largely satisfied with their response.

Reviewer #4: The authors carried out extensive studies ranging from biochemistry to animal models. Their results consistently demonstrated that US1 proteins from several alpha-herpesviruses, such as PRV and HSV-1, suppress cGAS-mediated interferon signaling by physical interaction with cGAS, therefore reducing dsDNA binding and cGAMP production. The authors further identified peptide sequences in the US1 proteins that adminstered the inhibition. AlphaFold models show the peptides bind to the cGAS catalytic domain. Deletion of US1 from viral genomes greatly elevated interferon signaling in infected cells and mice. Collectively, the authors showed that US1 antagonizes cGAS-mediate signaling, highlighting a promising target for the development of broad-spectrum alpha-herpesvirus therapeutics.

**Part II – Major Issues: Key Experiments Required for Acceptance**

Reviewer #1: (No Response)

Reviewer #2: (No Response)

Reviewer #3: (No Response)

Reviewer #4: The authors’ revision addressed the reviewers’ comments and most of my concerns.

**Part III – Minor Issues: Editorial and Data Presentation Modifications**

Reviewer #1: (No Response)

Reviewer #2: (No Response)

Reviewer #3: (No Response)

Reviewer #4: As a later comer, I do have a few minor comments to make.

1. In Figure 3F lane 3: SUMO-Flag-pcGAS+PRV, the SUMO-Flag-pcGAS amount is greatly reduced as shown in the IP: Flag/IB:Flag panel. This is peculiar as IP was conducted with mouse anti-Flag mAb. Further question: the authors can purify both PRV US1 and pcGAS from E. coli, the pull-down assay can be better carried out using purified proteins and Coomassie blue staining. Is there any reason co-IP was used?

2. Similarly, Figure S7E lanes 2&3. While the inputs only contain SUMO-FLAG-pcGAS or SUMO-FLAG-pcGAS+PRV US1, the IP: Flag/IB: Flag panel clearly blotted a band with lower molecular weight marked as “pcGASCT”. If proteolytic cleavage took place during co-IP, the Flag tag remains with the pcGASNT, i.e., pcGASCT should not be picked up by anti-Flag. The authors should reexamine the identity of the band.

3. For AlphaFold modeling in Figure 3A and Figure S12F-H, pLDDT score should be provided to better understand the reliability of prediction. “discontinuous” (line 231) is confusing. Do the authors mean “discrete”? It would be easier to state the starting and ending residue numbers of the two helices, or highlight them in Figure S2. In Figure 3A peptide A and peptide B appear to form a hairpin. It is unclear if they adopt a similar configuration in Figure S12F-H prediction. The author may consider expanding lines 456-457 (unmarked version) in Discussion.

4. EMSA protocol is confusing: 50 μM of ISD45 was mixed with cGAS at molar rations of 1:0, 1:2, …, 1:32. At molar ratio of 1:32, cGAS concentration is 1.6 mM, ~ 60 – 70 mg/ml for the protein. It is very difficult, if not impossible, to achieve such high concentration. The authors should reexamine the DNA and protein concentrations.

PLOS authors have the option to publish the peer review history of their article (what does this mean? ). If published, this will include your full peer review and any attached files.

**Do you want your identity to be public for this peer review?** For information about this choice, including consent withdrawal, please see our Privacy Policy .

Reviewer #1: No

Reviewer #2: No

Reviewer #3: No

Reviewer #4: **Yes: ** Qian Yin

---

## [Editor Report · Acceptance letter]

Dear Dr. Li,

We are delighted to inform you that your manuscript, "Alpha-herpesvirus US1 interacts with cGAS to suppress type I IFN responses and antiviral defense," has been formally accepted for publication in PLOS Pathogens.

Best regards,

Sumita Bhaduri-McIntosh

Editor-in-Chief

PLOS Pathogens

orcid.org/0000-0003-2946-9497

Michael Malim

Editor-in-Chief

PLOS Pathogens

orcid.org/0000-0002-7699-2064